# Understanding Transformers via $N$-gram Statistics

**Timothy Nguyen**
Google DeepMind
timothycnguyen@google.com

## Abstract

Transformer based large-language models (LLMs) display extreme proficiency with language yet a precise understanding of how they work remains elusive. One way of demystifying transformer predictions would be to describe how they depend on their context in terms of simple template functions. This paper takes a first step in this direction by considering families of functions (i.e. rules) formed out of simple $N$-gram based statistics of the training data. By studying how well these rulesets approximate transformer predictions, we obtain a variety of novel discoveries: a simple method to detect overfitting during training without using a holdout set, a quantitative measure of how transformers progress from learning simple to more complex statistical rules over the course of training, a model-variance criterion governing when transformer predictions tend to be described by $N$-gram rules, and insights into how well transformers can be approximated by $N$-gram rulesets in the limit where these rulesets become increasingly complex. In this latter direction, we find that for 79% and 68% of LLM next-token distributions on TinyStories and Wikipedia, respectively, their top-1 predictions agree with those provided by our $N$-gram rulesets.

## 1 Introduction

This paper is an attempt to answer the following

**Question:** *How does a transformer-based large language model (LLM) make use of its context when predicting the next token?*

Our approach proceeds via studying the statistical properties of training data. This is perhaps the most natural place to start even though it is not exhaustive (e.g. it does not include in-context learning [5]). The reasons to understand LLM behavior in terms of the statistics of their training data are plenty. First, the functional form of how LLMs use their training data is not well-understood (though there has been progress on understanding memorization [22, 6]). Second, the over-reliance of LLMs on training data statistics leads to brittleness (e.g. the "reversal curse" [3]) and the perpetuation of dataset biases [12]. Understanding the nature of this statistical dependence can lead to improved and more informed dataset curation and training methods. Finally, in various scenarios, the performance of LLMs on downstream tasks are found to be correlated with frequency of relevant training data [26, 10, 16, 17]. A better understanding of this phenomenon would allow better steering of models towards desired performance levels.

We can think of the complexity of an LLM next token prediction (regarded as a probability distribution over tokens) along two axes: form and selection. Form refers to the functional form of the prediction as a function of the context, e.g. whether the prediction is some explicit function of associated training data statistics (see Figure 1). Selection refers to which functional form, chosen from a set of functional templates, suitably describes the transformer prediction (supposing the choice set is sufficiently rich). As a first nontrivial step, one might hope that an approximate model for an LLM is that each of its next token predictions can be roughly described by simple statistical rules from

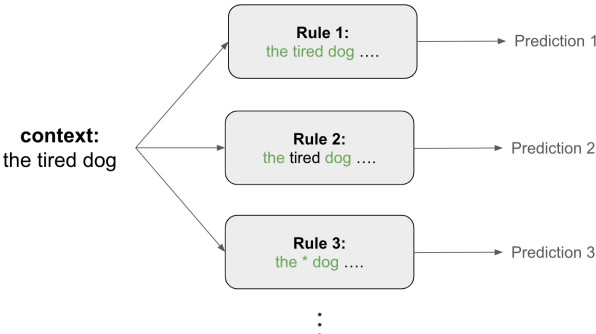

Figure 1: **Illustration of rule approximation.** Given a context, different $N$-gram based rules formed out of the context will yield different next-token predictive distributions. In the above example, the context consists of three tokens. The first rule uses all three tokens of the context and makes a prediction based on the corresponding 4-gram rule derived from the training data; the second rule uses only the first and last tokens to form a corresponding 3-gram rule (and so the next token "slept" will be assigned less weight than the first rule since the "tired" token is ignored); and the third rule makes a prediction using the $N$-gram statistics obtained from aggregating over three token contexts from the training data where the second token is arbitrary (i.e. the second token is marginalized). Given a list of such rules, one can ask which rule's predictive distribution best matches that of the transformer.

the context (simple form) even if the mechanism for its rule selection remains hidden (complex selection)[1]. This paper is an attempt to see how far this perspective can be pushed, and fortuitously we obtain additional insights for understanding LLM behavior along the way. The statistical rules we consider, which are based on $N$-grams, are defined in Section 4, with Figure 1 showing some examples.

We perform our main investigations on the TinyStories [11] dataset, with supporting experiments on Wikipedia to confirm our results remain robust at larger scales. The use of TinyStories is for practical reasons: its small size makes training models and aggregating $N$-gram statistics computationally efficient, yet it is complex enough to capture basic natural language statistics (those occurring in simple children's stories).

Below is a summary of our observations and contributions:

1. (Approximation Criterion) We observe that next token LLM predictions tend to be well-approximated by $N$-gram rules when the predictions have low variance across different training runs[2]. In particular, this includes predictions conditioned on contexts with sufficiently high count in the training data. (Section 5)

2. (Curriculum Learning Dynamics) By grouping our $N$-gram rulesets in terms of complexity (as measured by the amount of context they use), we discover the various ways in which the learning dynamics of LLMs implement a statistical type of curriculum learning, in which easier rules are eventually supplanted by more complex ones. (Section 6.1)

3. (Overfitting Criterion) Based on our analysis of approximating LLM predictions by $N$-gram rules, we propose a simple and novel procedure for detecting overfitting of LLMs during training. The procedure makes no use of holdout data and it makes quantatively precise the intuition that overfitting corresponds to a model memorizing long context at the expense being able to generalize through making use of subcontext. (Section 6.2)

4. (Approximation Strength) We study how well LLM predictions can be approximated by our $N$-gram rulesets, noting that significant gains in top1-accuracy occur as we increase ruleset

---

[1]It is important to emphasize that we seek a *descriptive* approximation of a transformer, rather than an *explanatory* one. A description merely requires that we can provide a post-hoc, per-instance approximation of transformer predictions in terms of an available rule; an explanation means we provide reasons for and thus can predict in advance why and when a particular rule approximates transformer predictions. Hence, we make the distinction between form (description) and selection (explanation).

[2]Different runs have different dataset shuffles.

complexity and diversity, whereby we achieve up to 79% top1-accuracy on TinyStories (Tables 2 and 5). We also visually ground these approximations with concrete examples (Figure 5), which may form the basis for dataset attribution methods in future work. Corresponding experiments on Wikipedia are shown in the Appendix. (Section 7)

We also open source our training datasets and related $N$-gram statistics so that others can verify and build upon our work.[3]

## 2   Related Work

Rule extraction methods for neural networks have been studied in quite different settings, e.g. [15, 21]. Some recent works have performed $N$-gram analyses for large-language models in the setting of in-context learning [1, 23] and associative recall [2]. The "infini-gram" model [19] compares LLM predictions with the single $N$-gram rule given by retrieving the largest possible matching context from the training data. Our work uses shorter but more sophisticated $N$-gram rules. In [28], an approach to understanding how LLMs process $N$-grams is carried out at the level of individual neurons. This complements our dataset-based work, which treat models as a black box. See also [27] which studies how transformers can represent $N$-gram models. In [9], the evolution of the type of $N$-gram statistics that transformers learn during training is analyzed in the setting of synthetic Markov chain data, in contrast to our natural language setting. Other works studying the learning trajectory of language models include [7, 8]. There is a large literature on building more sophisticated $N$-gram models, e.g. [18, 13]. Such models could have been incorporated into our set of rules, but for simplicity we choose not to include them.

## 3   Experimental Setup

We train standard decoder-only transformer models on the TinyStories [11] dataset (480M tokens) consisting of children's stories synthetically generated from GPT-4 using templated prompts. The value of this dataset lies in its linguistic simplicity, whereby it is possible to model language well on the dataset using very small models. Unless stated otherwise, our experiments use a 160M parameter model trained for 4 epochs, which achieves a loss of around 1.11 nats on the validation set. We train for 4 epochs since we use learning rate warmup and cosine learning rate decay and we want to ensure all datapoints receive updates with a high learning rate (this way all $N$-gram statistics have a fair chance of being learned during training). For overfitting experiments in Section 6.2, we train a 1.4B model for 10 epochs. In the Appendix, we include additional corresponding experiments on Wikipedia (from MassiveText [25]) with a single epoch of training in order to validate that our results are of a general nature and extend to more complex datasets. For a fixed dataset, the only source of randomness among different runs are different dataset shuffles. Full experimental details are described in the Appendix.

## 4   $N$-Gram Rules

The attention layer within a transformer is in essence a soft context-selection mechanism. The $N$-gram rules we consider will be loosely modeled on this mechanism. Namely, given a context we will form a derived context in which each token will either be kept, discarded, or marginalized, which is meant to mimic positive attention, no attention, and semantic invariance[4], respectively. More formally, we proceed as follows:

Given a regular expression[5] $\alpha$, all contexts from the training data can be retrieved which match the regular expression. This allows us to define a corresponding rule that defines for us a distribution

---

[3]https://github.com/google-deepmind/transformer_ngrams

[4]For instance, the next token distribution for the context "... the tired dog" may be insensitive to replacing "tired" with "brown" or "furry". Statistics which thus marginalize over all extant substitutions for "tired" yield a crude but generally applicable way of capturing semantic invariance. One can imagine an attention mechanism for which there is a many-to-one mapping of keys to a particular value that might implement semantic invariance.

[5]Our regular expressions operate on tokens not string characters, since our contexts are formed out of sequences of tokens.

over tokens $t$:

$$R_\alpha(t) = \frac{\#\{\alpha t\}}{\#\{\alpha *\}} \tag{1}$$

where the numerator and denominator are the counts for the $N$-grams from the training data matching the concatenated regular expressions $\alpha t$ and $\alpha *$, respectively, where $*$ is wildcard (single) character match[6]. (Thus the $N$-grams in the numerator end with $t$ while those in the denominator can end with any token.) Observe that the next-token predictions of a vanilla $N$-gram model are obtained by letting $R_\alpha(t)$ vary over all ordinary token sequences $\alpha$ of length $N-1$.

Given $\sigma$, a symbol from the the alphabet $\{*, -, +\}$, consider the following operation which maps a token $t$ to a regular expression:

$$S_\sigma(t) = \begin{cases} t & \sigma = + \\ * & \sigma = * \\ \epsilon & \sigma = - \end{cases} \tag{2}$$

where $\epsilon$ is the empty regular expression. Given now a sequence $\sigma = \sigma_{-N} \cdots \sigma_{-2}\sigma_{-1}$, define $S_\sigma$ on a sequence of tokens $C = C_{-N} \cdots C_{-2}C_{-1}$ by tokenwise application of (2) and concatenation[7]:

$$S_\sigma(C) = S_{\sigma_{-N}}(C_{-N}) \cdots S_{\sigma_{-2}}(C_{-2})S_{\sigma_{-1}}(C_{-1}). \tag{3}$$

Thus (3) defines a regular expression which we can think of as fuzzy matching for a subset of a context $C$ (the fuzziness arising from the presence of wildcard matches). For notational convenience, we assume $\sigma$ is left padded with $-$ symbols, so that we can define $S_\sigma(C)$ for $\text{len}(\sigma) < \text{len}(C)$. Finally, define

$$R_\sigma(t|C) = R_{S_\sigma(C)}(t) \tag{4}$$

for $C$ with $\text{len}(C) \geq \text{len}(\sigma)$. The collection of (4) for various $\sigma$ defines our $N$-gram rules under consideration[8]. Each such rule is a function which maps a context $C$ to a next token distribution. We refer to $S_\sigma(C)$ as the rule context for $R_\sigma(t|C)$.

As concrete examples, let $\sigma = + - *+$. If $C = C_{-5}C_{-4}C_{-3}C_{-2}C_{-1}$, then $S_\sigma(C) = C_{-4} * C_{-1}$ and

$$R_{+-*+}(t|C) = \frac{\#\{C_{-4} * C_{-1}t\}}{\#\{C_{-4} * C_{-1}*\}} \tag{5}$$

is a rule which yields a next token distribution based on a particular combination of 4-gram model statistics: it retrieves all three token contexts in the training data whose first token is $C_{-4}$ and last token is $C_{-1}$ and marginalizes over the second token. Likewise, the rules

$$R_{++--}(t|C) = \frac{\#\{C_{-4}C_{-3}t\}}{\#\{C_{-4}C_{-3}*\}} \qquad R_{++**} = \frac{\#\{C_{-4}C_{-3}**t\}}{\#\{C_{-4}C_{-3}***\}} \tag{6}$$

are respectively a trigram model with context $C_{-4}C_{-3}$ (all other tokens receiving a $-$ are dropped) and a model which uses four tokens of context but marginalizes over the two most recent ones.

When $\sigma$ consists of all $+$ symbols, we get vanilla $N$-gram rules derived from the suffix of $C$. When $\sigma$ consists of $\pm$ symbols, we get vanilla $N$-gram rules derived from subsets of $C$. Varying the length and the entries of $\sigma$ yields the following rulesets[9]:

$$\mathcal{R}_M^{\text{suffix}} = \{R_\sigma(t|\cdot) : |\sigma| \leq M, \sigma_i = + \text{ for all i}\} \tag{7}$$

$$\mathcal{R}_M^{\text{subgram}} = \{R_\sigma(t|\cdot) : |\sigma| \leq M, \sigma_i = \pm \text{ for all i}\} \tag{8}$$

$$\mathcal{R}_M^{\text{all}} = \{R_\sigma(t|\cdot) : |\sigma| \leq M\}. \tag{9}$$

The parameter $M$ controls how much of the context is being used for the rules.

---

[6]We use $*$ (i.e. glob notation) instead of the standard . symbol for readability purposes.

[7]The empty regular expression does nothing under concatenation and does not contribute to the length of the resulting sequence.

[8]For $\sigma = \emptyset$, we define $R_\sigma$ to be the unigram distribution.

[9]There is some redundancy among the $\sigma$'s in terms of the rules they determine: for instance, in between any two $+$ consecutive symbols, permuting the order of $-$ and * will determine the same rule. Also in practice, we can assume the first entry of $\sigma$ is a $+$ since marginalizing the first token is equivalent to reducing the context length. From this, it follows that the number of distinct rules in $\mathcal{R}_M^{\text{all}}$ is $2, 5, 13, 34, 89, 233, 378$, for $M = 1, \ldots, 7$, respectively.

Table 1: **Terminology associated to a context** $C$. Here $\mathcal{R}$ is some reference ruleset under consideration. The superscript on $p^{(i)}(t|C)$ is meant to denote the predictions of the $i$th model. In Section 5, we consider rules that are fixed across model runs (where we have five models) whereas elsewhere we will only have a single model (and thus optimal rules will be model specific).

| | |
|---|---|
| *optimal rule distance*: the minimum (possibly averaged over runs) distance between LLM predictions and rule predictions | $\min\limits_{r \in \mathcal{R}} \text{avg}_i d(p^{(i)}(t|C), p_r(t|C))$ |
| *optimal rule*: a rule achieving the optimal distance | $\underset{r \in \mathcal{R}}{\text{argmin}} \, \text{avg}_i d(p^{(i)}(t|C), p_r(t|C))$ |
| *model variance*: the average of the pairwise distance between LLM predictive distributions over different runs | $\underset{\substack{i,j \\ \text{distinct runs}}}{\text{avg}} \, d(p^{(i)}(t|C), p^{(j)}(t|C))$ |

## 5   Approximating Transformer Predictions with Rules

Let $p(t|C)$ denote the next-token distribution of an LLM conditioned on the context $C$ and for notational similarly, write $p_r(t|C)$ for $r(t|C)$, where $r$ is one of the rules defined in Section 4. We wish to measure how similar these distributions are (higher similarity corresponds to a better rule description). To that end, we use the variational distance to measure the difference of two distributions (we discuss our choice and others in the Appendix):

$$d(p, q) = \frac{1}{2} \sum_\alpha |p_\alpha - q_\alpha|, \tag{10}$$

where the summation is over the vocabulary index (i.e. the components of the probability vectors). Since variational distance may be lacking in concrete interpretability, we will sometimes use top1-accuracy to measure similarity, defined by

$$\text{top-1-acc}(p, q) = \frac{|\text{argmax}(p) \cap \text{argmax}(q)|}{|\text{argmax}(p) \cup \text{argmax}(q)|} \tag{11}$$

(in general, the argmax of a probability distribution is a set due to potential ties among maximal probabilities). When the argmaxes in (11) are singletons, top1-accuracy just measures agreement between greedy predictions.

Given a context $C$, we want to understand how $d(p(t|C), p_r(t|C))$ varies with different rules $r$ and in particular if it can be made small. To that end, we introduce some terminology:

We are interested in determining the optimal rule $p_r(t|C)$ (as defined in Table 1) and if it has small optimal rule distance then we regard the rule as being a good description of the corresponding transformer predictions.[10] As a first step, note there is a distinguished rule

$$p_{\text{full}}(t|C) = \frac{\#\{Ct\}}{\#\{C*\}} \tag{12}$$

whose rule context is the full unmodified context $C$.[11] This is because (roughly) the language-modeling objective aims to make $p(t|C)$ similar to $p_{\text{full}}(t|C)$.[12] All other rules in our rulesets are "subleading" in that they drop or marginalize over tokens in the context $C$. Our goal is to quantify which rules, either (12) or subleading ones, are optimal rules and what their optimal rule distances are.

One of our main findings is an *approximation criterion*: contexts that have low model-variance tend to have low optimal rule distance. In particular, this includes contexts with sufficiently high frequency in the training data. The latter situation is to be expected: the more often a context $C$ occurs in the

---

[10]In practice, we will only have one model available and our optimal rules are computed per-context and per-model. In this section, we have available five models from five runs for use in computing optimal quantities.

[11]That is, $p_{\text{full}}(t|C)$ is the invocation of the rule corresponding to $\sigma = + \cdots + \in \mathcal{R}_{|C|}^{\text{suffix}}$ applied to $C$.

[12]See Section C for additional discussion.

training data, the more the minimization of the cross entropy loss objective encourages the network to make predictions close to $p_{\text{full}}(t|C)$.

The novel aspect of our approximation criterion is the *sufficiency* of low model-variance situation even in cases when the context is rare.[13] We present the case of 7-gram contexts in Figure 2 to corroborate the approximation criterion, with additional examples relegated to the Appendix. We sample around six-thousand 7-grams from the training data, sampling from logarithmically spaced buckets based on counts, and plot various relations between counts, model variances, and rule distances. For simplicity, we consider the ruleset $\mathcal{R} = \mathcal{R}_7^{\text{suffix}}$ to limit the number of rules under consideration. Our analysis of Figure 2 can be summarized as follows:

Plot (a) shows how with increasing count of the number occurrences of the context $C$ in the training data, LLM predictions become nearer to $p_{\text{full}}(t|C)$, which in this case, is the vanilla 8-gram rule. Nevertheless, for all but the highest of counts, we have a large spread of distances: even for unique 7-gram contexts, some predictions are well-approximated by $p_{\text{full}}(t|C)$ while others are close to having disjoint-support (distance equal to 1). Plot (c) also shows that while model variance decreases with count of the context (as expected) we have a large spread of model variances for contexts with intermediate or low counts. Since the contexts whose predictions have high model variance can be regarded as "noise", one can ask whether those contexts with low model variance have some structure. Plots (b) and (c) address this question. While for (b), we see that the 8gram-rule has a large spread when plotted against model variance, there is a significant reduction in outliers in (d) when the y-axis is the optimal distance to rules in $\mathcal{R}_7^{\text{suffix}}$. Concretely, the transition from (b) to (d) is a way of visualizing LLMs performing back-off, whereby LLMs rely on statistics from subsets of the context. Moreover, the lower left portion of (d) is a manifestation of our approximation criterion: contexts that yield consistent predictions across model runs (i.e. sufficiently low model variance) are indicative of rule-like behavior (in this case, good approximation with a suffix $N$-gram rule formed out of the context).

We believe our approximation criterion and its corresponding analyses have significance beyond the experiments carried out here since they (i) highlight that naive count-based statistics do not provide the strongest signal in terms of how LLMs leverage dataset statistics (since as Figure 2(a) shows, high count can still have high model variance) (ii) suggest that LLM predictions that have low-variance are likely the ones that are amenable to description (or even explanation) by some underlying dataset statistic (with high-variance predictions being regarded as noise). We leave a more systematic exploration of (ii) to future work.

## 6   Learning Dynamics

### 6.1   Curriculum Learning

We can track how well LLM predictions are described by our $N$-gram rules over the course of training by tracking optimal rule distance as a function of train step. Here optimal rule distance is defined as in Table 1 with $\mathcal{R}$ any of the rulesets (7)-(9), and we will measure how optimal rule distances vary with maximum context length $M$ (the resulting analyses are similar for "all", "subgram", and "suffix" rules so we show our analysis for "all").

Figure 3 summarizes our results. Early in training, LLM predictions acquire structure and thus become approximable by rule predictors. However, with further training, LLM predictions eventually diverge from simpler rules (small context length) while continuing to increase in similarity with more complex rules (larger context length). Moreover, the rightmost plot of Figure 3 shows that top1-acc$(p_{\text{gt}}(t|C), p_r(t|C))$ increases over the course of training for optimal $r \in \mathcal{R}_M^{\text{all}}$ (for $M > 1$), where $p_{\text{gt}}$ is the ground-truth distribution regarded as a one-hot distribution, showing that the rule selection improves with time. Altogether, this shows that LLMs undergo a curriculum style learning, in which their predictions gradually move away from simpler rules to more complex and effective rules.

---

[13]Necessity is a given. Predictions which have high variance cannot be well approximated by a single model-independent rule. We use five runs in our analysis here since approximation by a rule that remains fixed across models yields a fortiori approximation by a per-model rule.

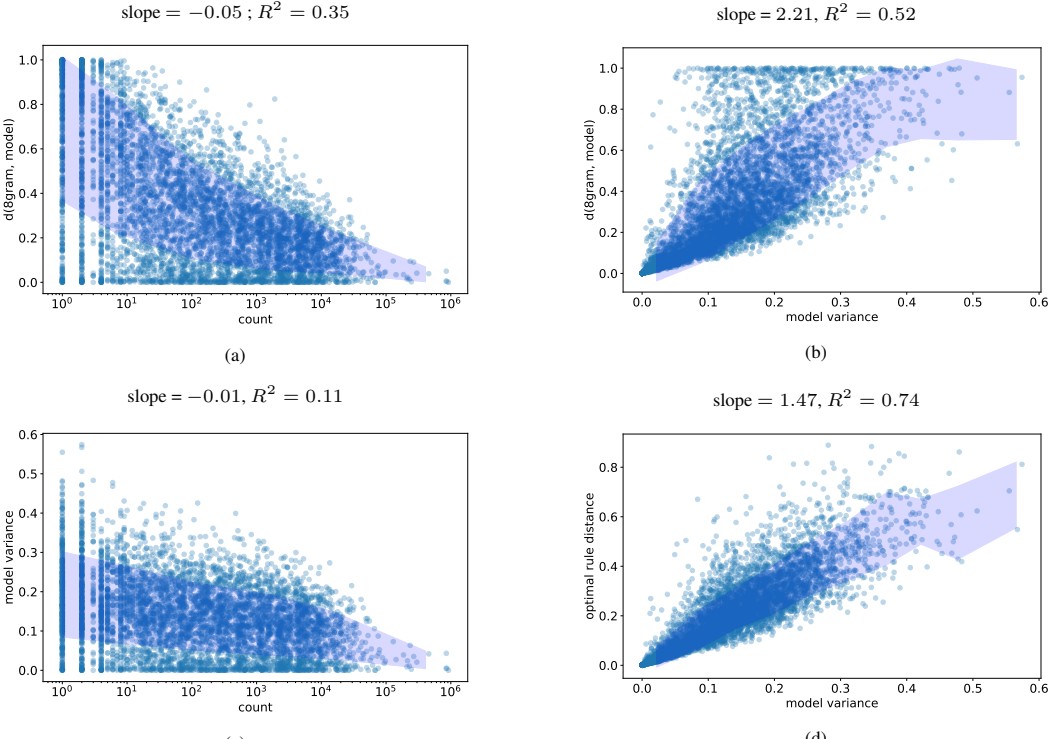

Figure 2: **TinyStories** 7-**grams**. Every point in the above plots represents a 7-gram context. Shaded regions are obtained by bucketing along the x-axis and computing one standard deviation within the mean along the y-axis. Slope and $R^2$ values of plots are with respect to the linear fit of the data. Optimal rule distances and model variances are computed with respect to five model runs. *(a)*: $d(p(t|C), p_{\text{full}}(t|C))$ vs count of $C$. *(b)*: $d(p(t|C), p_{\text{full}}(t|C))$ vs model variance. *(c)*: model variance vs count of $C$. *(d)*: similar to (b) but now the y-axis is optimal rule distance of the optimal rule from $\mathcal{R}_7^{\text{suffix}}$. Model size: 160M.

## 6.2 Early Stopping Criterion

Our investigations of approximating LLMs with rules given by limited contexts naturally lead us to consider LLMs with limited context. The latter have predictive distributions given by

$$p_n(t|C) = p(t|C_{-n} \cdots C_{-1}) \tag{13}$$

where $n$ is the maximum context length. In Figure 4, we plot the loss of an LLM trained to overfit (train loss decreases while validation loss increases) along with its limited context versions for $1 \leq n \leq 7$. For the limited context models with $n > 1$, we see that on *both* the train and validation set, the two respective loss curves track each other closely and both eventually go up. This suggests the following picture: an overfitting LLM is spending capacity to minimize train loss by memorizing the full context at the expense of using capacity to learn statistics of subcontext, i.e. the reduced context in (13). This manifests itself both during training (where subcontext arises from a subset of a larger memorized context) and during validation (where subcontext arises from the partial overlap between novel context and the train set).

Our discovery suggests a simple and computationally inexpensive early stopping criterion: during training, evaluate the transformer on train data consisting of short contexts and when this quantity begins increasing, stop training. Significantly, this method involves no holdout set and is a training dataset intrinsic criterion.

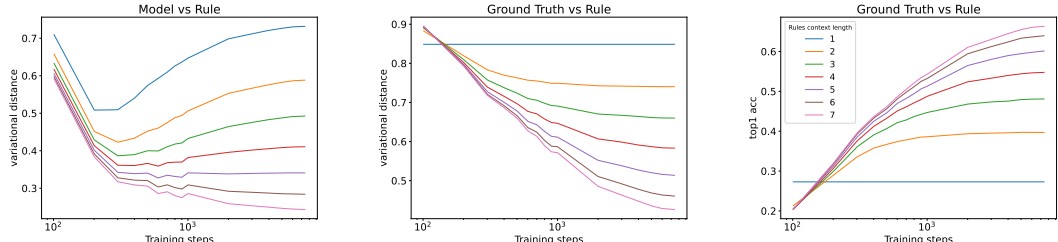

Figure 3: **Training Dynamics.** *Left:* Models reach their lowest distance to more complex rules later in training. For rules with four tokens of context or less, the variational distance eventually starts increasing later in training. For six and seven tokens of context, the variational distance continues to decrease. *Center & Right:* The optimal rule selected always has nonincreasing distance and nondecreasing top1-accuracy relative to the ground truth (interpreted as a one-hot distribution $p_{\text{gt}}$), despite distances to model predictions eventually increasing or plateauing for rules with less than six tokens of context. This shows that the optimal rule selection is improving with additional training even if the optimal rule distance with respect to model predictions is not improving. (One can imagine the rule predictions as a mesh in probability space, with LLM predictions navigating this space through training. The distance to the mesh may plateau but which rule is closest can continue to change.) Model size: 160M.

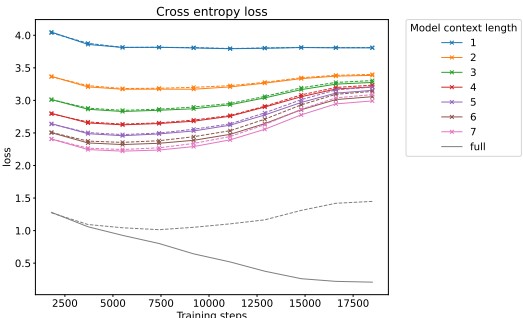

Figure 4: **Overfitting Detection.** We plot both train loss (solid lines) and validation loss (dashed lines) for the full transformer and limited context length transformers (the latter are marked with an "x" for emphasis) on TinyStories. Unlike the full transformer which overfits, those with limited context length have train and validation loss curves closely following each other. Model size: 1.4B.

## 7 Rule Peformance

Finally, addressing our main question from the introduction, we track how well our rulesets describe LLM predictions (in the sense of Section 5) as a whole at inference time. Here, the utility of our $N$-gram rules defined in Section 4 becomes apparent, since on a holdout set, there will be novel contexts and being able to drop or marginalize context tokens aid in being able to retrieve or aggregate corresponding training dataset statistics. In Table 2, we show the average top1-accuracy between the optimal rule from our various rulesets and LLM predictions on 100 random stories from the validation set. Here, we include as baseline backoff$_M$, the single rule given by the predictive model which performs "stupid backoff" [4] using $M$ tokens of context.[14]

We see significant gains in accuracy at large $M$ when adding additional types of rules. In the end, we are able to obtain 78% top1-accuracy between the per-prediction optimal rule and the LLM predictions, averaged over all tokens. This is perhaps a remarkably high figure, considering that the top1 accuracy of the model with respect to the ground truth on the validation set is 69.6%. At minimum, we have provided a precise quantification of structure in LLM next-token predictions: they are often matched (as measured by top token prediction) by some simple $N$-gram rule derived from

---

[14]That is, $p_{\text{backoff}_M}(t|C) = p_{\text{full}}(t|C_{-k}\cdots C_{-1})$ where $k \leq M$ is the largest value for which $C_{-k}\cdots C_{-1}$ occurs in the training data.

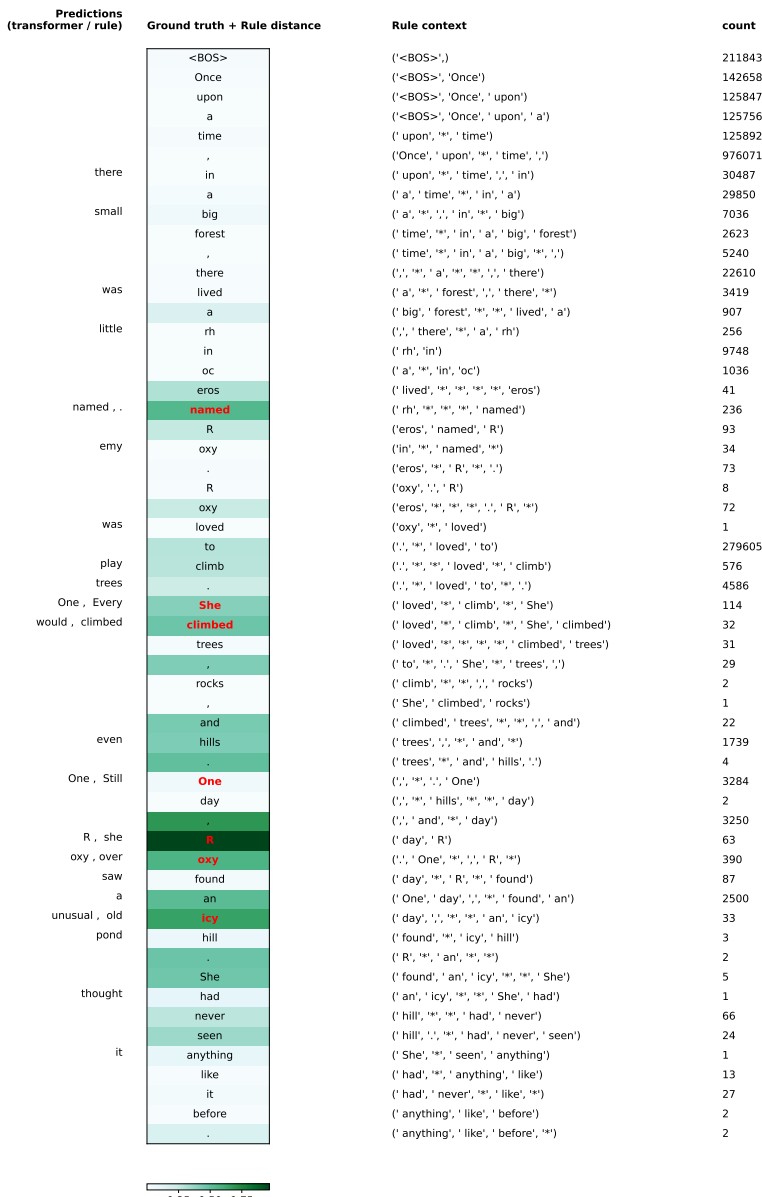

Figure 5: **Rule selection for a TinyStories validation sequence.** The above is a sequence from a heldout story. In the second and third columns are the ground truth, token by token, along with the rule context (as defined in Section 4) associated to the optimal rule from $\mathcal{R}_7^{\text{all}}$. The heatmap indicates the variational distance between optimal rule and LLM next token distributions at the given token. The first column shows at most two tokens, which are chosen as follows: If the LLM top-1 prediction disagrees with the ground truth, the LLM prediction is shown. If in addition, the rule selected makes a different top-1 prediction from the transformer, that token is shown as the second token and the corresponding ground truth token is colored red. Thus red tokens are precisely the locations of disagreement between LLM and optimal rule greedy predictions. The last column shows the number of contexts supporting the optimal rule. Model size: 160M.

the training data. See Section D.1 for some supplementary analysis. Using the $L^\infty$ distance instead of the variational distance gives us a slightly higher result of 79%, see Table 5.

To ground our rule optimization procedure, we provide Figure 5 which shows side-by-side how LLM predictions compare with ground truth and optimal rule predictions in an example heldout story.

Table 2: **Top-1 accuracy of optimal rule**. We look at the average top1-accuracy of the optimal rule versus LLM predictions for rules of varying strength and maximum rule context length $M$. We compute this average over each token prediction from 100 random validation stories (around 22K tokens total). Model size: 160M.

| ruleset / context length | 1 | 2 | 3 | 4 | 5 | 6 | 7 |
|---|---|---|---|---|---|---|---|
| $\mathcal{R}_M^{\text{all}}$ | 30.0 | 44.8 | 54.3 | 62.4 | 68.8 | 74.0 | 78.0 |
| $\mathcal{R}_M^{\text{subgram}}$ | 30.0 | 44.5 | 53.1 | 60.0 | 64.8 | 68.5 | 71.1 |
| $\mathcal{R}_M^{\text{suffix}}$ | 30.0 | 44.4 | 52.2 | 57.9 | 61.5 | 63.8 | 65.6 |
| $\text{backoff}_M$ | 30.0 | 42.5 | 48.7 | 52.7 | 54.6 | 55.9 | 56.7 |

For instance, for the target token "`climb`" in "`...  Roxy loved to climb`", both the LLM and optimal rule $R_\alpha$ predict "`play`", where $\alpha$ = "`.  * loved to`". For target token "`climb`" in "`... She climbed`", the LLM predicts "`would`" whereas the ground truth and $R_\alpha$ predict "`climb`", where $\alpha$ = "`loved to climb * She`". In general, optimal rules provide the closest statistical match from the training data to the given LLM predictive distributions (from amongst our rulesets), and their top1-predictions can agree or disagree agree (as indicated by target token color). Additional examples, including those from Wikipedia, are shown in Section D. For interpretability purposes, we re-emphasize that our optimal rules currently only provide descriptions, not explanations. We leave the possibility of the latter for future work.

## 8    Conclusions and Limitations

Our work provides quantitative measures of how well the predictions of transformer-based LLMs are described (i.e. approximated) by simple $N$-gram rules. Such rules were motivated by the simplest token-level operations applied to the context (keep, ignore, or marginalize). The results we obtained in Section 7 imply that, at least on simple datasets like TinyStories and Wikipedia, LLM predictions contain much quantifiable structure insofar that they often can be described in terms of our simple statistical rules. Along the way, we also obtained novel discoveries into the statistical nature of overfitting, the occurrence of curriculum learning, and the relation between model-variance and approximability by $N$-gram rules. Altogether then, our work provides various avenues of progress in understanding how simple dataset statistics are reflected in LLM behavior.

On the other hand, it is intuitively clear that current state-of-the-art LLMs go well beyond invoking $N$-gram rules. A typical request to perform a nontrivial task (e.g. "Write a thirty line poem about mathematics that rhymes") requires a high-level conceptual understanding of language that goes beyond simple literal token-level associations between the context and the training data that we consider here. Nevertheless, one can speculate that an analogue of our work could still apply: in general, an LLM might be performing some high-level rule application, whereby statistics formed out of distributional categories [24] instead of individual tokens are leveraged from the context. Formulating a correct and parsimonious set of rules, if it is at all possible, would be a nontrivial challenge to overcome and one which we leave to future work. Addressing such a challenge and being able to promote the descriptive approximations provided here to explanatory ones would provide a next step towards understanding how LLMs work.

## Acknowledgements

The author would like to especially thank Senthooran Rajamanoharan for numerous conversations and an exceptionally discerning eye, which greatly improved the paper from earlier drafts. The author also thanks Jonathan Hale, Marcus Hutter, Matthew McGill, Nick Roy, Avraham Ruderman, and Daniel Tanis for helpful feedback and discussions. Finally, the author thanks Frank Perbet and Daniel Tanis for engineering support.

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

## A Choice of Distance Measure

We choose variational distance since it is a symmetric and bounded distance function (unlike the KL divergence). Symmetry means we do not have to make a choice between computing the distance between model predictions and rule predictions or vice versa. Boundedness ensures that when we measure average distance across tokens, large outliers do not dominate the average. In fact, for the KL divergence, since $KL(p||q)$ is infinite when $p > 0$ whenever $q = 0$, were we to use KL divergence, we would have to set $p$ equal to rule predictions and $q$ equal to model predictions (since rule predictions are typically sparse). To avoid such constraints and potential pathologies, we choose the variational distance. Another possible choice is the Jensen-Shannon distance, but we found it gives similar results to variational distance.

It is worth noting that while the $L^\infty$ metric

$$d_{L^\infty}(p, q) = \max_\alpha |p_\alpha - q_\alpha| \tag{14}$$

often gives slightly better results for the rule-approximation analysis of Section 7, it has a failure mode when comparing two very high entropy distributions. If $p$ and $q$ are two distributions such that $p_\alpha$ and $q_\alpha$ are all small, then their $L^\infty$ distance will be small even though their variational distance can be large. Thus, the $L^\infty$ distance is not suitable for the curriculum learning analysis analysis in Section 6.1, since at initialization when the model makes uniform predictions on tokens, it will have low $L^\infty$ distance with many bad $N$-gram rules that are high entropy. Hence, for the sake of using a consistent metric in the main body of the paper, we use the variational distance when comparing probability distributions, although as the numbers of Section D show, for well-trained models the $L^\infty$ distance often yields better rule approximation.

## B Additional Experimental Details

Our transformer architecture and training procedure is based on that of Chinchilla [14]. The architecture hyperparameters are as follows:

Table 3: Model specifications.

| Model | Layers | Number Heads | $d_{key}/d_{value}$ | $d_{model}$ |
|-------|--------|--------------|---------------------|-------------|
| 160M  | 12     | 16           | 64                  | 896         |
| 420M  | 12     | 16           | 128                 | 1536        |
| 1.4B  | 24     | 16           | 128                 | 2048        |

We use a linear learning rate warmup of 1000 steps up to a maximum value of $2 \times 10^{-4}$ and then use a cosine learning rate decay. We use weighted Adam optimizer [20] with weight decay $10^{-4}$. Our models are trained using TPU accelerators. The 160M and 420M models use 16 TPU accelerators while the 1.4B models use 64 TPU accelerators per run. We use a batch size of 128 sequences with each sequence consisting of 2048 tokens.

Our training datasets (TinyStories and MassiveText Wikipedia) are prepared as follows. After tokenizing the individual documents (stories for TinyStories and articles for Wikipedia), we concatenate them all into one long sequence, with each document separated by the ⟨BOS⟩ token[15]. The full sequence is then subdivided into contiguous sequences of length 2048 (with padding as needed) and then shuffled to form a static dataset of shuffled sequences. We refer to the previous procedure as "chunking". Crucially, observe that chunking results in most sequences not starting with the ⟨BOS⟩ token (hence a model will be trained to predict the next token conditioned on incomplete contexts, as desired).

For TinyStories experiments, we train 160M models for 4 epochs except for the overfitting experiments in Section 6.2 where we train 1.4B models for 10 epochs. We use the train and validation splits provided by HuggingFace[16]. For Wikipedia experiments, we train a 1.4B model for a single epoch.

---

[15]Attention masks are used so that tokens only attend to those from the same document

[16]Available at https://huggingface.co/datasets/roneneldan/TinyStories

We have train and validation splits based on using choosing random sets of disjoint documents. Our Wikipedia train set has 4.4B tokens. In places where we perform several training runs (Section 5), the only source of variance (randomness) among the runs are different dataset shuffles. The only exception to the above is in Section D.3, where all model sizes are trained for 1 epoch (for TinyStories, we observed overfitting of the 1.4B model around 4 epochs so we switch to 1 epoch for all model sizes for a fair comparison).

Our tokenizer[17] uses byte-pair encoding trained on MassiveText with a vocabulary size of 32,678.

### B.1  $N$-gram statistics

The computation of $N$-gram statistics of the training data is formed after chunking (as described above), so that they correspond to the $N$-gram statistics seen by models during training. In particular, tokens which are contiguous in a story but separated by the chunking will not contribute to the $N$-gram statistics. We used a distributed map-reduce system to tabulate $N$-gram counts in the most naive manner. Using sliding windows of size $N$ and aggregating across train documents, we are able to compute $N$-gram counts for all occurring $N$-grams and store them in SQL databases. (We ignore those invalid $N$-grams where $\langle \text{BOS} \rangle$ occurs not as the first token). Note that the number of rows of such $N$-gram databases is bounded by at most the size of the training corpus times $N$.

As an aside, we note that for the analysis in Section 6.1, we used our static $N$-gram rules computed from the entire training data. We do not compute statistics based on the training dataset seen up to the point in training. However, for the purposes of our analysis, this distinction is immaterial (and in practice, the distinction between two sets of statistics will, for the dominant $N$-grams, be small with sufficiently large batch size).

## C  Additional Approximation Criterion Analyses

We provide additional commentary and experimental settings for our analysis in Section 5.

### C.1  Full-context vs Subcontext

As noted in footnote 12, there is usually a mismatch between the contexts that $N$-gram rules and LLMs receive during training: the latter can receive very long contexts (up to one less than the number of tokens in a document) while the former typically receives very short contexts (in our case, up to 7 tokens). Concretely, while a bigram model is trained on consecutive pairs of tokens $(c, t)$, an LLM is rarely trained so as to optimize $p(t|c)$. Indeed, given a training sequence $x$, only the target for the first token of $x$ has context consisting of a single token; the other targets will have more tokens of context accordingly. Thus, it is unclear how well LLM predictions $p(t|c)$ should match bigram rule predictions as $c$ varies over the vocabulary set, since LLMs almost always receive $c$ within a much larger context. More generally, it is unclear how well $p(t|C)$ matches $p_{\text{full}}(t|C)$. Nevertheless, because in practice LLMs learn how to use context effectively, LLMs manage to learn $p(t|C)$ despite being optimized for $p(t|\tilde{C})$ with $\tilde{C}$ a context containing $C$ as a suffix.

As a measure of how much training context "dilutes" the LLM ability to learn the bigram distribution, in Figure 6 we plot the distance between LLM predictions and the bigram rule for two LLMs: one trained in the usual fashion with full context and one trained with only one token of context (concretely, a token can only attend to itself in attention layers). In both cases, we have the same pattern of increased count leads to lower rule distance. However, the transformer trained with context length equal to one has much lower distances since it cannot learn anything else other than the bigram rule. The difference between the variational differences of the two models is thus a measure of the dilution an LLM has in learning a bigram rule owing to receiving surrounding context.

As an aside, we note how for both models, a context with low count has difficulty being learned. In this way, one can regard the inability to learn rules for low count contexts as being due to a failure of optimization, something that could be addressed in the future by improved optimization methods.

---

[17]Trained using https://github.com/google/sentencepiece

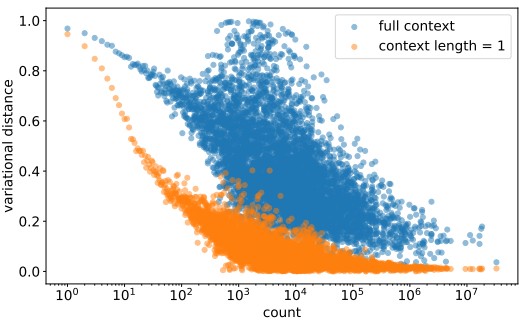

Figure 6: **Comparison with TinyStories bigram model.** We evaluate transformer models (trained with either full context or context length equal to one) on all 22.8K distinct unigrams of TinyStories and record the corresponding variational distance with the bigram rule. Grouping unigrams based on count and averaging the variational distances result in the above scatterplots. Model size: 160M.

## C.2 TinyStories Unigram Context

We repeat Figure 2 for the simplest case of unigram context. In this case, there is only one rule (the bigram rule) and so there are only three plots to consider.

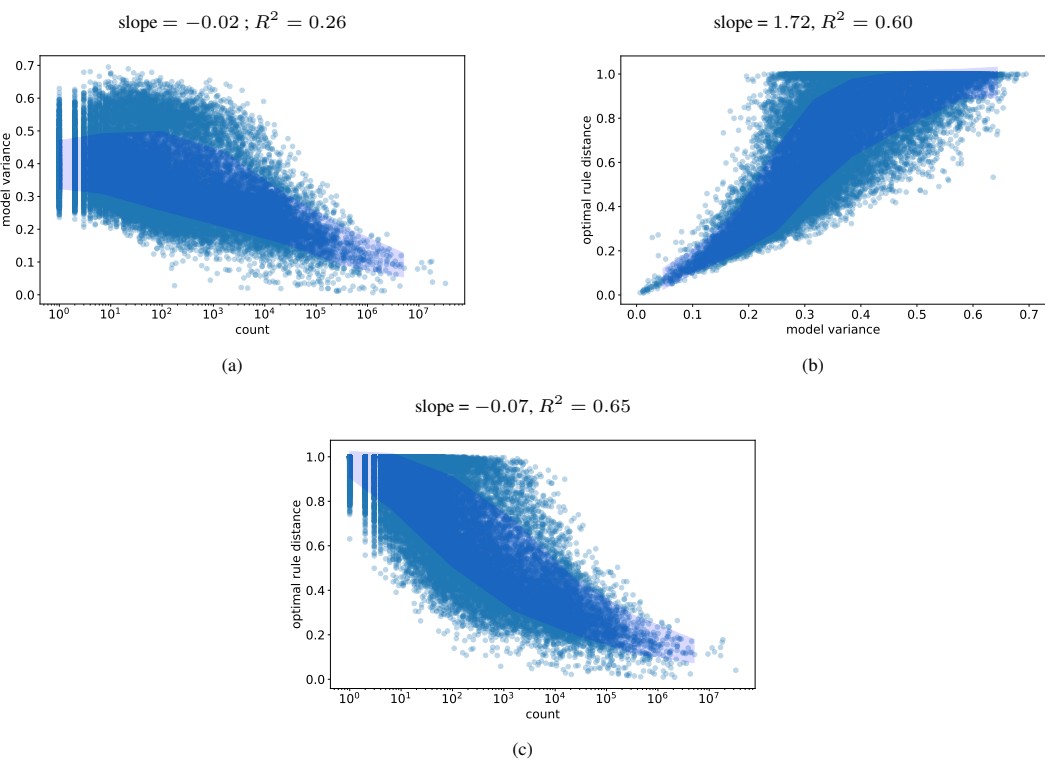

Figure 7: **TinyStories 1-grams**. Every point in the above plots represents a 1-gram context (all 22.8K from TinyStories). Shaded regions are plots obtained by bucketing along the x-axis and computing one standard deviation within the mean along the y-axis. Slope and $R^2$ values of plots are with respect to the linear fit of the data given by their axes. Optimal rule distances and model variances are computed with respect to five model runs. *(a)*: model variance vs count of $C$ *(b)*: $d(p(t|C), p_{\text{full}}(t|C))$ vs model variance *(c)*: $d(p(t|C), p_{\text{full}}(t|C))$ vs count. Model size: 160M.

## C.3 Tinystories Bigram Context

Next, consider the case when there are two tokens of context. To get a more fine-grained analysis, we consider the case of full-context bigrams, i.e. those starting with the $\langle\text{BOS}\rangle$ token. This is because such bigrams do not appear within a larger context and so a transformer's corresponding predictions are more fair to compare with those of $N$-gram models (both are trained using equal contexts). Conveniently, there are only 691 full-context bigrams in this case and so we do not have to randomly sample a subset.

We will consider the ruleset $\mathcal{R}_2^{\text{all}}$ for which there are three relevant $N$-gram rules of interest: one which uses the entire bigram of context (a trigram model), one which uses only the last token (a bigram model), and one which uses only the first token (the next token distribution of $\langle\text{BOS}\rangle$).[18] We will refer to these as the trigram, bigram, and $\langle\text{BOS}\rangle$ rule respectively.

We plot an analog of Figure 2 for full-context bigrams in Figure 8. As with the analysis for Figure 2(a), a large count leads to good approximation with $p_{\text{full}}(t|C)$, which is the vanilla trigram rule at present. However lower counts lead to a spread in approximability (some low counts have high error while others have low ones). In (c), we plot a variation in which the $x$-axis is the maximum of the count of $C$ and the unigram $C_{-1}$. What the poor fit in (c) indicates is that whether a prediction is well-described by a rule is not a simply determined by whether a subcontext of $C$ occurs often. Given the small number of rules, we now color code the optimal rule of each full-context bigram (as indicated by the legend in (b)). In passing from (b) to (d), we see how the outliers in the upper left of (b) move towards the bottom once the large distance from the trigram model is replaced with the optimal rule distance. These are bigrams whose rules are well approximated by bigram or $\langle\text{BOS}\rangle$ rules and are misspecified when trying to be approximated by the trigram rule. In accord with our Approximation Criterion, contexts with low model variance are well approximated by $N$-gram rules (the lower left of (d)).

## C.4 Wikipedia 6-gram contexts

We plot the analog of Figure 2 in Figure 9 but for contexts consisting of 6-grams from Wikipedia. We also subsample as before, from logarithmically spaced buckets, for a total of around 6.8K total contexts. We get nearly identical behavior as with TinyStories. Our Approximation Criterion is thus not specific to small datasets like TinyStories.

---

[18]It turns out that the $\langle\text{BOS}\rangle*$ rule (given by $R_{+*}$) in $\mathcal{R}_2^{\text{all}}$ never occurs as an optimal rule for full-context bigrams and so can be ignored in this example.

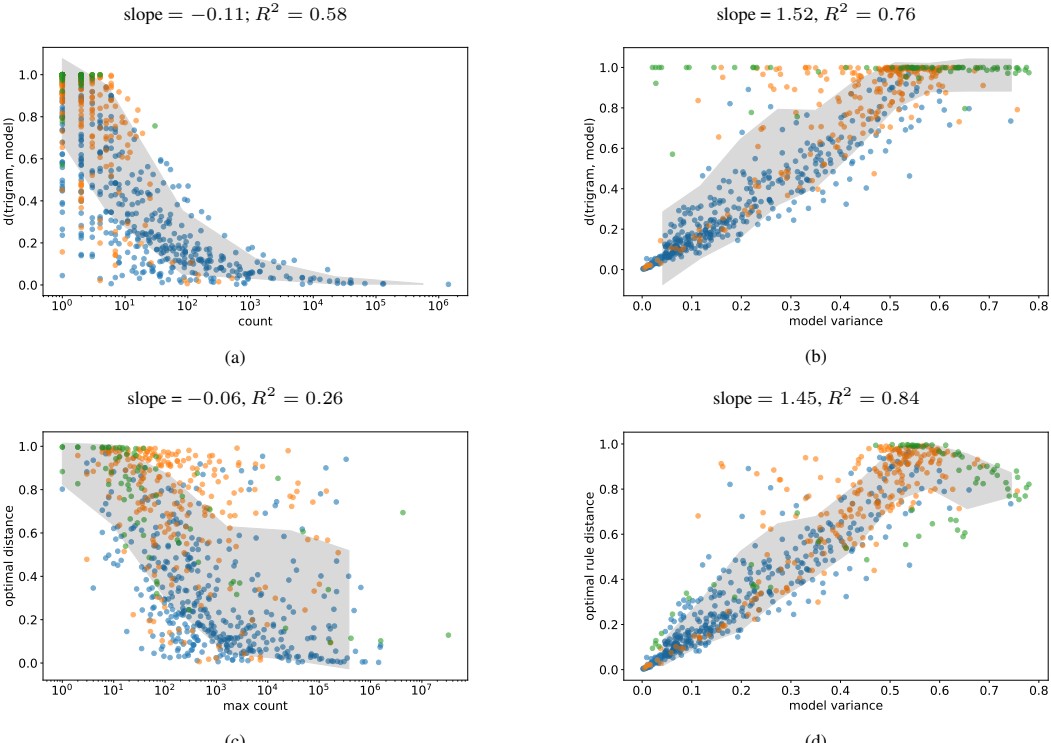

Figure 8: **TinyStories full-context bigrams**. Every point in the above plots represents a full-context bigram $C$ from among the 691 distinct ones in TinyStories. Points are colored by which $N$-gram rule is the optimal rule, among those in $\mathcal{R}_2^{\text{all}}$, for transformer prediction given $C$. Shaded regions are plots obtained by bucketing along the x-axis and computing one standard deviation within the mean along the y-axis. *(a):* $d(p(\cdot|C), p_{\text{trigram}}(\cdot|C))$ vs count of $C$. *(b):* $d(p(\cdot|C), p_{\text{trigram}}(\cdot|C))$ vs model variance. *(c):* Optimal rule distance vs the greater of the bigram count of $C$ and the unigram count of $C_{-1}$. *(d):* Similar to upper right but now the y-axis is optimal rule distance. Five model runs are used to compute optimal rule distance and model variance. Model size: 160M.

# D   Rule Performance: Additional Analysis and Examples

## D.1   TinyStories

We supplement Table 2 with Table 4 to show how optimal rule distances decrease with increasing rule strength. This is to preclude a trivial situation in which by having sufficiently many rules (say a one-hot distribution for every vocabulary token), one can have a ruleset that for any model prediction always returns an optimal rule with 100% top-1 accuracy! Such coarse rules will not in general yield small optimal distances however[19] and our variational distances decreasing in Table 4 shows that our rulesets are truly better approximating the predictions with increasing strength.

We also include the analog of Tables 2 and (4 but with the variational distance replaced with the $L^\infty$ distance in Tables 5 and 6. We see that in fact the top-1 accuracy numbers are slightly better in this case.

## D.2   Wikipedia

We present analogous results for Section 7 using a 1.4B model trained on Wikipedia. In Table 7 we present the analogue of Table 2 on ten holdout Wikipedia chunks (a total of $10 \times 2048$ tokens). The

---

[19]Both the variational and $L^\infty$ distances between a one-hot distribution and a distribution which is uniform on $n$ tokens are at least $\frac{n-1}{n}$. Thus, whenever an LLM has at least two roughly valid options, we expect a one-hot distribution to be at least of distance 0.5 from the LLM prediction.

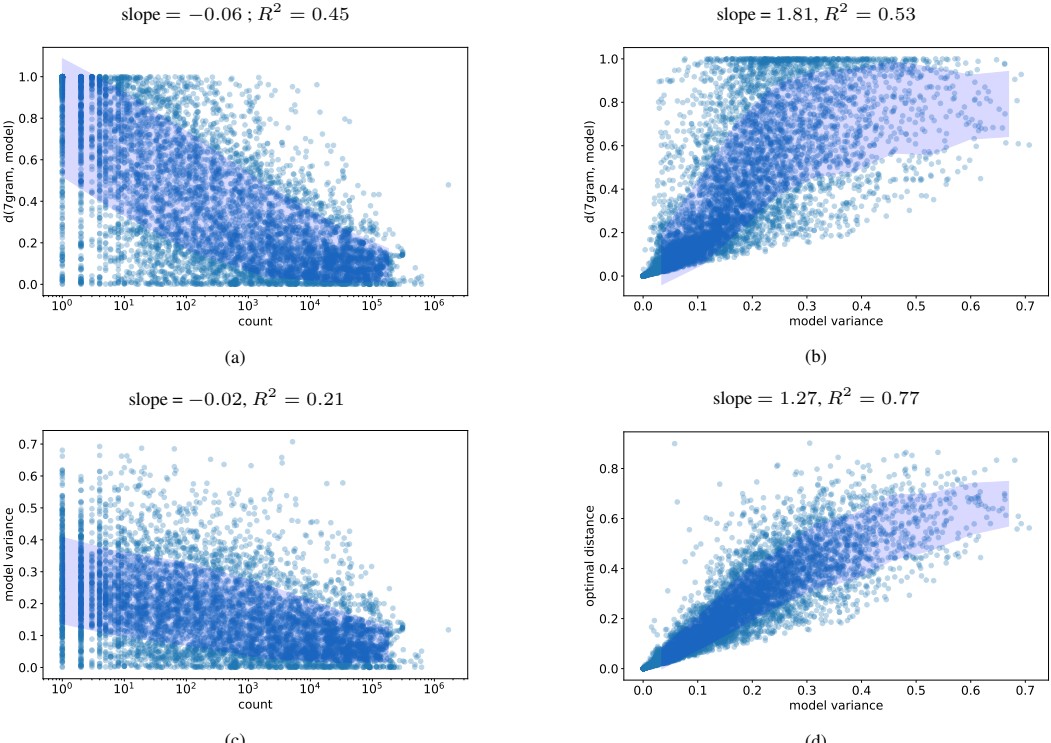

Figure 9: **Wikipedia 6-grams**. Every point in the above plots represents a 6-gram context. Shaded regions are plots obtained by bucketing along the x-axis and computing one standard deviation within the mean along the y-axis. Slope and $R^2$ values of plots are with respect to the linear fit of the data given by their axes. Optimal rule distances and model variances are computed with respect to five model runs. *(a)*: $d(p(t|C), p_{\text{full}}(t|C))$ vs count of $C$. *(b)*: $d(p(t|C), p_{\text{full}}(t|C))$ vs model variance. *(c)*: model variance vs count of $C$. *(d)*: similar to (b) but now the y-axis is optimal rule distance of the optimal rule from $\mathcal{R}_6^{\text{suffix}}$. Model size: 160M.

Table 4: **Optimal rule distance (TinyStories, variational distance)**. We look at the average optimal rule distance with LLM predictions for rules of varying strength and maximum context length $M$. We compute this average over each token prediction from 100 random TinyStories validation stories (around 22K tokens total). Model size: 160M.

| ruleset / context length | 1 | 2 | 3 | 4 | 5 | 6 | 7 |
|---|---|---|---|---|---|---|---|
| $\mathcal{R}_M^{\text{all}}$ | 0.738 | 0.597 | 0.507 | 0.434 | 0.37 | 0.316 | 0.274 |
| $\mathcal{R}_M^{\text{subgram}}$ | 0.738 | 0.598 | 0.513 | 0.449 | 0.399 | 0.362 | 0.335 |
| $\mathcal{R}_M^{\text{suffix}}$ | 0.738 | 0.598 | 0.519 | 0.465 | 0.425 | 0.399 | 0.381 |
| $\text{backoff}_M$ | 0.738 | 0.606 | 0.539 | 0.503 | 0.482 | 0.472 | 0.466 |

top-1 accuracy when using optimal rules from $\mathcal{R}_7^{\text{all}}$ and the $L^\infty$ distance for rule selection is 67.7%. As with TinyStories, we see significant gains in accuracy when we increase rule strength. Achieving the number 67.7% (versus the corresponding 78.9% number for TinyStories from Table 5), perhaps a surprisingly a high score, is the result of two competing factors: on the one-hand, Wikipedia is a more complex dataset (which makes prediction harder), while on the other hand, the training data has more $N$-grams and thus more rules. Our model achieves 54.9% top-1 accuracy on the 10 holdout Wikipedia chunks (Table 12) which is substantially lower than the top-1 accuracy of the optimal rule.

Table 5: **Top-1 accuracy of optimal rule (TinyStories, $L^\infty$ distance)**. The analogue of Table 2 but with $L^\infty$ distance instead of variational distance for selecting the optimal rule. Model size: 160M.

| ruleset / context length | 1 | 2 | 3 | 4 | 5 | 6 | 7 |
|---|---|---|---|---|---|---|---|
| $\mathcal{R}_M^{\text{all}}$ | 30.0 | 45.1 | 55.0 | 63.2 | 69.6 | 75.0 | 78.9 |
| $\mathcal{R}_M^{\text{subgram}}$ | 30.0 | 44.7 | 53.5 | 60.4 | 65.1 | 68.5 | 71.2 |
| $\mathcal{R}_M^{\text{suffix}}$ | 30.0 | 44.5 | 52.6 | 58.0 | 61.4 | 63.6 | 65.2 |
| $\text{backoff}_M$ | 30.0 | 42.5 | 48.7 | 52.7 | 54.6 | 55.9 | 56.7 |

Table 6: **Optimal rule distance (TinyStories, $L^\infty$ distance)**. The analogue of Table 4 but with $L^\infty$ distance instead of variational distance. Model size: 160M.

| ruleset / context length | 1 | 2 | 3 | 4 | 5 | 6 | 7 |
|---|---|---|---|---|---|---|---|
| $\mathcal{R}_M^{\text{all}}$ | 0.557 | 0.443 | 0.371 | 0.309 | 0.253 | 0.207 | 0.170 |
| $\mathcal{R}_M^{\text{subgram}}$ | 0.557 | 0.444 | 0.376 | 0.322 | 0.28 | 0.248 | 0.225 |
| $\mathcal{R}_M^{\text{suffix}}$ | 0.557 | 0.446 | 0.383 | 0.338 | 0.305 | 0.282 | 0.267 |
| $\text{backoff}_M$ | 0.557 | 0.456 | 0.407 | 0.385 | 0.378 | 0.379 | 0.382 |

## D.3 Model Scaling

In this section, we present results about how our rule approximation changes with model size. For both TinyStories and Wikipedia, we train models of size 160M, 420M, and 1.4B for one epoch[20].

In Tables 11 and 12, there is a clear trend towards improved model performance with scale: we obtain lower cross entropy loss, higher top-1 accuracy, and lower model distance to the ground truth distribution regarded as a one-hot distribution[21]. Note that for the latter, distance from model predictions to the ground truth distribution is the same with respect to the variational or $L^\infty$ distance and is given by $1 - p$ where $p$ is the probability assigned by the model to the ground truth.

On the other hand, entries in Tables 13-20 measuring the changes in rule approximation with scale are much more modest (i.e. are more stable) by comparison. They also show mixed results, with $L^\infty$ results worsening but some variational distance results slightly improving with scale (for large context length rules). We leave a more thorough investigation of how model scale affects approximability by $N$-gram rules to future work.

---

[20]Since the 1.4B starts showing signs of overfitting around 4 epochs on TinyStories, we train models for 1 epoch in this section unlike 4 epochs elsewhere.

[21]It would be more proper to aggregate statistics over the holdout set to compute a ground truth distribution that takes into the account the relative frequencies of the next token given the context. However, since most contexts will be unique, this more refined computation will not affect the corresponding result much.

Table 7: **Top-1 accuracy of optimal rule (Wikipedia, $L^\infty$ distance)**. We look at the average top-1 accuracy between optimal rule and LLM predictions for rules of varying strength and maximum context length. We compute this average over each token prediction from 10 holdout Wikipedia sequences each consisting of 2048 tokens. Model size: 1.4B.

| ruleset / context length | 1 | 2 | 3 | 4 | 5 | 6 | 7 |
|---|---|---|---|---|---|---|---|
| $\mathcal{R}_M^{\text{all}}$ | 24.3 | 39.7 | 49.8 | 55.5 | 60.3 | 64.3 | 67.7 |
| $\mathcal{R}_M^{\text{subgram}}$ | 24.3 | 39.2 | 48.3 | 52.7 | 55.5 | 57.6 | 59.0 |
| $\mathcal{R}_M^{\text{suffix}}$ | 24.3 | 38.8 | 47.1 | 50.3 | 51.5 | 52.3 | 52.7 |
| $\text{backoff}_M$ | 24.3 | 36.8 | 42.9 | 43.9 | 43.7 | 43.8 | 43.9 |

Table 8: **Optimal rule distance (Wikipedia, $L^\infty$ distance)**. We look at the average $L^\infty$ distance between optimal rule and LLM predictions for rules of varying strength and maximum context length. We compute this average over each token prediction from 10 holdout Wikipedia sequences each consisting of 2048 tokens. Model size: 1.4B.

| ruleset / context length | 1 | 2 | 3 | 4 | 5 | 6 | 7 |
|---|---|---|---|---|---|---|---|
| $\mathcal{R}_M^{\text{all}}$ | 0.48 | 0.369 | 0.29 | 0.237 | 0.202 | 0.173 | 0.150 |
| $\mathcal{R}_M^{\text{subgram}}$ | 0.48 | 0.371 | 0.295 | 0.249 | 0.226 | 0.209 | 0.198 |
| $\mathcal{R}_M^{\text{suffix}}$ | 0.48 | 0.375 | 0.303 | 0.265 | 0.250 | 0.242 | 0.238 |
| $\text{backoff}_M$ | 0.48 | 0.394 | 0.359 | 0.368 | 0.387 | 0.398 | 0.405 |

Table 9: **Top-1 accuracy of optimal rule (Wikipedia, variational distance)**. We look at the average top-1 accuracy between optimal rule and LLM predictions for rules of varying strength and maximum context length. We compute this average over each token prediction from 10 holdout Wikipedia sequences each consisting of 2048 tokens. Model size: 1.4B.

| ruleset / context length | 1 | 2 | 3 | 4 | 5 | 6 | 7 |
|---|---|---|---|---|---|---|---|
| $\mathcal{R}_M^{\text{all}}$ | 24.3 | 39.2 | 48.8 | 54.4 | 58.5 | 61.9 | 65.0 |
| $\mathcal{R}_M^{\text{subgram}}$ | 24.3 | 39.0 | 47.9 | 52.3 | 54.9 | 56.8 | 58.3 |
| $\mathcal{R}_M^{\text{suffix}}$ | 24.3 | 38.7 | 46.9 | 50.3 | 51.5 | 52.4 | 52.8 |
| $\text{backoff}_M$ | 24.3 | 36.8 | 42.9 | 43.9 | 43.7 | 43.8 | 43.9 |

Table 10: **Optimal rule distance (Wikipedia, variational distance)**. We look at the average variational distance between optimal rule and LLM predictions for rules of varying strength and maximum context length. We compute this average over each token prediction from 10 holdout Wikipedia sequences each consisting of 2048 tokens. Model size: 1.4B.

| ruleset / context length | 1 | 2 | 3 | 4 | 5 | 6 | 7 |
|---|---|---|---|---|---|---|---|
| $\mathcal{R}_M^{\text{all}}$ | 0.768 | 0.635 | 0.543 | 0.484 | 0.446 | 0.413 | 0.388 |
| $\mathcal{R}_M^{\text{subgram}}$ | 0.768 | 0.636 | 0.549 | 0.498 | 0.472 | 0.453 | 0.440 |
| $\mathcal{R}_M^{\text{suffix}}$ | 0.768 | 0.638 | 0.556 | 0.513 | 0.497 | 0.488 | 0.483 |
| $\text{backoff}_M$ | 0.768 | 0.656 | 0.609 | 0.597 | 0.598 | 0.599 | 0.600 |

Table 11: **TinyStories metrics.** How average cross entropy loss, top-1 accuracy, and model distance to the ground truth scale with model size on a holdout set of 100 stories.

| model size | eval loss | eval acc | eval distance |
|---|---|---|---|
| 160M | 1.43 | 63.2 | 0.485 |
| 420M | 1.28 | 65.9 | 0.452 |
| 1.4B | 1.22 | 66.9 | 0.439 |

Table 12: **Wikipedia metrics.** How average cross entropy loss, top-1 accuracy, and model distance to the ground truth scale with model size on a holdout set of 10 Wikipedia chunks.

| model size | eval loss | eval acc | eval distance |
|---|---|---|---|
| 160M | 2.63 | 50.0 | 0.617 |
| 420M | 2.43 | 52.3 | 0.590 |
| 1.4B | 2.26 | 54.9 | 0.562 |

Table 13: **Top-1 accuracy of optimal rule with model scale (TinyStories, variational distance).** How top-1 accuracy of the optimal rule varies with model size and rule context length. Optimal rule from is selected from $\mathcal{R}_7^{\text{all}}$ using variational distance.

| model size / context length | 1 | 2 | 3 | 4 | 5 | 6 | 7 |
|---|---|---|---|---|---|---|---|
| 160M | 31.4 | 47.1 | 56.7 | 64.3 | 69.8 | 74.5 | 77.9 |
| 420M | 30.7 | 45.9 | 55.5 | 63.3 | 69.2 | 74.1 | 77.9 |
| 1.4B | 30.5 | 45.9 | 55.3 | 63.4 | 69.6 | 74.3 | 78.2 |

Table 14: **Optimal rule distance with model scale (TinyStories, variational distance).** How optimal rule distance varies with model size and rule context length. Optimal rule is selected from $\mathcal{R}_7^{\text{all}}$ using variational distance.

| model size / context length | 1 | 2 | 3 | 4 | 5 | 6 | 7 |
|---|---|---|---|---|---|---|---|
| 160M | 0.692 | 0.545 | 0.46 | 0.398 | 0.347 | 0.306 | 0.275 |
| 420M | 0.711 | 0.566 | 0.479 | 0.411 | 0.355 | 0.310 | 0.274 |
| 1.4B | 0.718 | 0.574 | 0.486 | 0.417 | 0.359 | 0.311 | 0.274 |

Table 15: **Top-1 accuracy of optimal rule with model scale (TinyStories, $L^\infty$ distance).** How top-1 accuracy of the optimal rule varies with model size and rule context length. Optimal rule is selected from $\mathcal{R}_7^{\text{all}}$ using $L^\infty$ distance.

| model size / context length | 1 | 2 | 3 | 4 | 5 | 6 | 7 |
|---|---|---|---|---|---|---|---|
| 160M | 31.4 | 47.5 | 57.5 | 65.2 | 71.4 | 76.1 | 79.6 |
| 420M | 30.7 | 46.3 | 56.3 | 64.2 | 70.5 | 75.3 | 79.2 |
| 1.4B | 30.5 | 46.2 | 56.0 | 64.1 | 70.5 | 75.3 | 79.3 |

Table 16: **Optimal rule distance with model scale (TinyStories, $L^\infty$ distance).** How optimal rule distance varies with model size and rule context length. Optimal rule is selected from $\mathcal{R}_7^{\text{all}}$ using $L^\infty$ distance.

| model size / context length | 1 | 2 | 3 | 4 | 5 | 6 | 7 |
|---|---|---|---|---|---|---|---|
| 160M | 0.482 | 0.368 | 0.301 | 0.248 | 0.204 | 0.169 | 0.141 |
| 420M | 0.511 | 0.398 | 0.329 | 0.272 | 0.223 | 0.184 | 0.153 |
| 1.4B | 0.522 | 0.408 | 0.338 | 0.280 | 0.230 | 0.189 | 0.157 |

Table 17: **Top-1 accuracy of optimal rule with model scale (Wikipedia, variational distance).** How top-1 accuracy of the optimal rule varies with model size and rule context length. Optimal rule is selected from $\mathcal{R}_7^{\text{all}}$ using variational distance.

| model size / context length | 1 | 2 | 3 | 4 | 5 | 6 | 7 |
|---|---|---|---|---|---|---|---|
| 160M | 25.9 | 40.8 | 49.5 | 54.2 | 57.7 | 61.0 | 63.8 |
| 420M | 24.7 | 39.8 | 49.0 | 54.2 | 58.1 | 61.6 | 64.3 |
| 1.4B | 24.3 | 39.2 | 48.8 | 54.4 | 58.5 | 61.9 | 65.0 |

Table 18: **Optimal rule distance with model scale (Wikipedia, variational distance)**. How optimal rule distance varies with model size and rule context length. Optimal rule is selected from $\mathcal{R}_7^{\text{all}}$ using variational distance.

| model size / context length | 1 | 2 | 3 | 4 | 5 | 6 | 7 |
|---|---|---|---|---|---|---|---|
| 160M | 0.737 | 0.606 | 0.527 | 0.477 | 0.445 | 0.418 | 0.397 |
| 420M | 0.753 | 0.621 | 0.534 | 0.480 | 0.445 | 0.414 | 0.391 |
| 1.4B | 0.768 | 0.635 | 0.543 | 0.484 | 0.446 | 0.413 | 0.388 |

Table 19: **Top-1 accuracy of optimal rule with model scale (Wikipedia, $L^\infty$ distance)**. How top-1 accuracy of the optimal rule varies with model size and rule context length. Optimal rule is selected from $\mathcal{R}_7^{\text{all}}$ using $L^\infty$ distance.

| model size / context length | 1 | 2 | 3 | 4 | 5 | 6 | 7 |
|---|---|---|---|---|---|---|---|
| 160M | 25.9 | 41.2 | 50.4 | 55.9 | 60.4 | 64.4 | 67.8 |
| 420M | 24.7 | 40.3 | 50.0 | 55.7 | 60.4 | 64.3 | 67.7 |
| 1.4B | 24.3 | 39.7 | 49.8 | 55.5 | 60.3 | 64.3 | 67.7 |

Table 20: **Optimal rule distance with model scale (Wikipedia, $L^\infty$ distance).** How optimal rule distance varies with model size and rule context length. Optimal rule is selected from $\mathcal{R}_7^{\text{all}}$ using $L^\infty$ distance.

| model size context length | 1 | 2 | 3 | 4 | 5 | 6 | 7 |
|---|---|---|---|---|---|---|---|
| 160M | 0.429 | 0.321 | 0.253 | 0.208 | 0.179 | 0.154 | 0.135 |
| 420M | 0.455 | 0.346 | 0.271 | 0.222 | 0.190 | 0.163 | 0.143 |
| 1.4B | 0.480 | 0.369 | 0.290 | 0.237 | 0.202 | 0.173 | 0.150 |

| Predictions (transformer / rule) | Ground truth + Rule distance | Rule context | count |
|---|---|---|---|
| | <BOS> | ('<BOS>',) | 2118438 |
| | Once | ('<BOS>', 'Once') | 1426581 |
| | upon | ('<BOS>', 'Once', ' upon') | 1258475 |
| | a | ('<BOS>', 'Once', ' upon', ' a') | 1257566 |
| | time | (' upon', ' a', ' time') | 1258881 |
| | , | ('Once', ' upon', ' a', ' time', ',') | 976049 |
| there | in | (' upon', ' a', ' time', ',', ' in') | 30487 |
| | a | (' a', ' time', ',', ' in', ' a') | 29850 |
| small | big | (' a', ' time', ',', ' in', ' a', ' big') | 7021 |
| | forest | (' time', ',', ' in', ' a', ' big', ' forest') | 2623 |
| | , | (' time', ',', ' in', ' a', ' big', ' forest', ',') | 2617 |
| | there | (' big', ' forest', ',', ' there') | 2725 |
| was | lived | (' big', ' forest', ',', ' there', ' lived') | 966 |
| | a | (' big', ' forest', ',', ' there', ' lived', ' a') | 907 |
| little | rh | (' big', ' rh') | 351 |
| | in | (' rh', 'in') | 9748 |
| | oc | (' a', ' rh', 'in', 'oc') | 1034 |
| | eros | (' lived', ' a', ' rh', 'in', 'oc', 'eros') | 41 |
| named , . | **named** | (' rh', 'in', 'oc', 'eros', ' named') | 236 |
| | R | ('eros', ' named', ' R') | 93 |
| emy | oxy | (' named', ' R', 'oxy') | 18 |
| | . | ('eros', ' named') | 237 |
| | R | ('oxy', '.', ' R') | 8 |
| | oxy | ('oxy', '.', ' R', 'oxy') | 8 |
| was | loved | ('oxy', ' loved') | 4 |
| | to | (' loved', ' to') | 546806 |
| play | climb | (' loved', ' to', ' climb') | 2524 |
| trees | . | (' loved', ' to', ' climb', '.') | 485 |
| One , He | **She** | (' loved', ' to', ' climb', '.', ' She') | 111 |
| would , climbed | **climbed** | (' loved', ' to', ' climb', '.', ' She', ' climbed') | 32 |
| | trees | (' climb', '.', ' She', ' climbed', ' trees') | 21 |
| | , | (' climb', '.', ' She', ' climbed', ' trees', ',') | 19 |
| | rocks | (' climb', '.', ' She', ' climbed', ' trees') | 21 |
| | , | (' She', ' climbed', ' rocks') | 1 |
| | and | (' trees', ',', ' rocks', ',', ' and') | 117 |
| even | hills | (',', ' and', ' hills') | 29 |
| | . | (' trees', ',', ' and', ' hills', '.') | 4 |
| One , Still | **One** | ('.', ' One') | 846242 |
| | day | (',', ' and', ' hills', '.', ' One', ' day') | 1 |
| | , | ('.', ' One', ' day', ',') | 717085 |
| R , she | **R** | (' day', ' R') | 63 |
| oxy , over | **oxy** | (' day', ',', ' R', 'oxy') | 17 |
| saw , was | **found** | ('oxy', ' found') | 2 |
| a | an | (' found', ' an') | 14301 |
| unusual , old | **icy** | (' found', ' an', ' icy') | 59 |
| pond , lake | **hill** | (' found', ' an', ' icy', ' hill') | 2 |
| | . | (' R', '.') | 6 |
| She , He | **She** | (' hill', '.', ' She') | 3214 |
| thought , was | **had** | (' icy', '.', ' She', ' had') | 6 |
| never , to | **never** | (' had', ' never') | 71696 |
| | seen | (' hill', '.', ' She', ' had', ' never', ' seen') | 11 |
| it | anything | (' She', ' never', ' seen', ' anything') | 1 |
| | like | (' had', ' seen', ' anything', ' like') | 11 |
| | it | (' She', ' had', ' never', ' seen', ' it') | 567 |
| | before | (' anything', ' like', ' before') | 2 |
| | . | (' anything', ' like', ' before', '.') | 2 |

0.25  0.50  0.75

Figure 10: **Rule selection for a TinyStories heldout sequence using $\mathcal{R}_7^{\mathrm{subgram}}$.** Analogous to Figure 5 but with optimal rule chosen from $\mathcal{R}_7^{\mathrm{subgram}}$ instead of $\mathcal{R}_7^{\mathrm{all}}$. Model size: 160M.

## D.4   Rule Selection

Here we supplement our example in Figure 5 by showing how the smaller rulesets $\mathcal{R}_7^{\mathrm{subgram}}$ and $\mathcal{R}_7^{\mathrm{suffix}}$ compare in Figures 10 and 11. As expected, the top1 accuracy between transformer predictions and optimal rule predictions decrease with smaller rulesets.

| Predictions (transformer / rule) | Ground truth + Rule distance | Rule context | count |
|---|---|---|---|
| | <BOS> | ('<BOS>',) | 2118438 |
| | Once | ('<BOS>', 'Once') | 1426581 |
| | upon | ('<BOS>', 'Once', ' upon') | 1258475 |
| | a | ('<BOS>', 'Once', ' upon', ' a') | 1257566 |
| | time | (' upon', ' a', ' time') | 1258881 |
| | , | ('Once', ' upon', ' a', ' time', ',') | 976049 |
| there | in | (' upon', ' a', ' time', ',', ' in') | 30487 |
| | a | (' a', ' time', ',', ' in', ' a') | 29850 |
| small | big | (' a', ' time', ',', ' in', ' a', ' big') | 7021 |
| | forest | (' time', ',', ' in', ' a', ' big', ' forest') | 2623 |
| | , | (' time', ',', ' in', ' a', ' big', ' forest', ',') | 2617 |
| | there | (' big', ' forest', ',', ' there') | 2725 |
| was | lived | (' big', ' forest', ',', ' there', ' lived') | 966 |
| | a | (' big', ' forest', ',', ' there', ' lived', ' a') | 907 |
| little | rh | (' lived', ' a', ' rh') | 41 |
| | in | (' rh', 'in') | 9748 |
| | oc | (' a', ' rh', 'in', 'oc') | 1034 |
| | eros | (' lived', ' a', ' rh', 'in', 'oc', 'eros') | 41 |
| named , . | named | (' rh', 'in', 'oc', 'eros', ' named') | 236 |
| | R | ('eros', ' named', ' R') | 93 |
| emy | oxy | (' named', ' R', 'oxy') | 18 |
| | . | (' named', ' R', 'oxy', '.') | 11 |
| R , She | R | ('oxy', '.', ' R') | 8 |
| | oxy | ('oxy', '.', ' R', 'oxy') | 8 |
| was | loved | ('oxy', ' loved') | 4 |
| | to | (' loved', ' to') | 546806 |
| play | climb | (' loved', ' to', ' climb') | 2524 |
| trees | . | (' loved', ' to', ' climb', '.') | 485 |
| One , He | She | (' loved', ' to', ' climb', '.', ' She') | 111 |
| would , climbed | climbed | (' loved', ' to', ' climb', '.', ' She', ' climbed') | 32 |
| | trees | (' climb', '.', ' She', ' climbed', ' trees') | 21 |
| | , | (' climb', '.', ' She', ' climbed', ' trees', ',') | 19 |
| | rocks | (' climbed', ' trees', ',', ' rocks') | 37 |
| | , | (' trees', ',', ' rocks', ',') | 148 |
| | and | (' trees', ',', ' rocks', ',', ' and') | 117 |
| even | hills | (',', ' and', ' hills') | 29 |
| | . | (' hills', '.') | 2155 |
| One , \n | One | ('.', ' One') | 846242 |
| | day | (',', ' and', ' hills', '.', ' One', ' day') | 1 |
| | , | ('.', ' One', ' day', ',') | 717085 |
| R , she | R | (',', ' R') | 3223 |
| oxy , emy | oxy | (' day', ',', ' R', 'oxy') | 17 |
| saw , was | found | ('oxy', ' found') | 2 |
| a | an | (' found', ' an') | 14301 |
| unusual , old | icy | (' found', ' an', ' icy') | 59 |
| pond , lake | hill | (' found', ' an', ' icy', ' hill') | 2 |
| | . | ('.',) | 32885210 |
| She , \n | She | (' hill', '.', ' She') | 3214 |
| thought , was | had | (' hill', '.', ' She', ' had') | 97 |
| never , a | never | (' had', ' never') | 71696 |
| | seen | (' hill', '.', ' She', ' had', ' never', ' seen') | 11 |
| it | anything | (' hill', '.', ' She', ' had', ' never', ' seen', ' anything') | 2 |
| | like | (' She', ' had', ' never', ' seen', ' anything', ' like') | 1621 |
| | it | (' She', ' had', ' never', ' seen', ' anything', ' like', ' it') | 1453 |
| | before | (' had', ' never', ' seen', ' anything', ' like', ' it', ' before') | 4529 |
| | . | (' anything', ' like', ' it', ' before', '.') | 3577 |

0.25  0.50  0.75

Figure 11: **Rule selection for a TinyStories heldout sequence using $\mathcal{R}_7^{\text{suffix}}$.** Analogous to Figure 5 but with optimal rule chosen from $\mathcal{R}_7^{\text{suffix}}$ instead of $\mathcal{R}_7^{\text{all}}$. Model size: 160M.

We also ground our rule approximation on Wikipedia by providing two concrete examples in Figures 12 and 13.

| Predictions (transformer / rule) | Ground truth + Rule distance | Rule context | count |
|---|---|---|---|
| | , | (',',) | 153786312 |
| | and | (',', ' and') | 13109555 |
| | the | (',', '*', ' the') | 6464695 |
| | front | (',', ' the', '*') | 9208196 |
| of | toes | (',', ' and', ' the', ' front', '*') | 900 |
| | are | (' and', '*', ' toes', ' are') | 90 |
| long , | partially | (',', ' and', '*', ' toes', ' are', '*') | 16 |
| web | joined | (' and', '*', '*', '*', '*', ' partially', '*') | 1516 |
| | at | (' the', '*', '*', ' are', '*', ' joined', '*') | 384 |
| | the | (' are', '*', ' joined', '*', ' the') | 470 |
| | base | (' are', '*', '*', ' at', '*', ' base') | 706 |
| | . | (' base', '.') | 32190 |
| The , \n | F | (' partially', '*', '*', '*', '*', '*', ' F') | 29 |
| oss , a | if | ('.', ' F', 'if') | 3007 |
| | teen | (' base', '*', '*', 'teen') | 10 |
| | species | (' base', '*', '*', 'teen', ' species') | 3 |
| | have | ('teen', ' species', ' have') | 151 |
| | been | ('teen', ' species', ' have', '*') | 151 |
| | recorded | ('teen', '*', '*', '*', ' recorded') | 158 |
| | in | (' species', '*', '*', ' recorded', '*') | 3796 |
| South , | Guyana | (' species', '*', '*', '*', '*', ' Guyana') | 74 |
| | . | (' species', '*', ' been', '*', '*', '*', '.') | 1915 |
| \n | \n | (' have', ' been', '*', ' in', '*', '.', '\n') | 2229 |
| \n | \n | (' Guyana', '.', '\n', '\n') | 810 |
| The | Blue | (' recorded', '*', '*', '*', '*', '*', 'Blue') | 50 |
| | - | ('Blue', '-') | 2562 |
| headed , w | and | ('Blue', '-', 'and') | 128 |
| | - | ('\n', 'Blue', '-', '*', '-') | 105 |
| | white | ('white',) | 71413 |
| col , " | swallow | ('-', '*', '*', '*', ' swallow') | 143 |
| | , | (' swallow', ',') | 2449 |
| T , | Py | ('white', ' swallow', '*', '*') | 36 |
| | g | ('white', ' swallow', '*', '*', '*') | 36 |
| | oc | (' swallow', '*', '*', '*', 'oc') | 48 |
| | hel | (' swallow', '*', '*', '*', 'oc', '*') | 48 |
| | id | (' Py', 'g', 'oc', '*', '*') | 32 |
| | on | (',', '*', 'g', '*', '*', 'id', 'on') | 27 |
| | cyan | (' Py', 'g', '*', '*', 'id', '*', '*') | 35 |
| | ole | ('on', ' cyan', 'ole') | 22 |
| | uc | ('on', ' cyan', '*', 'uc') | 22 |
| | a | (' cyan', '*', 'a') | 346 |
| ( | \n | ('ole', 'uc', '*', '*') | 3662 |
| \n | Black | (' cyan', '*', '*', '*', '*', 'Black') | 17 |
| | - | (' cyan', '*', '*', '\n', '*', '-') | 96 |
| and | coll | ('Black', '*', 'coll') | 95 |
| | ared | ('uc', 'a', '*', '*', '*', '*', 'ared') | 8 |
| | swallow | ('a', '*', '*', '*', 'ared', ' swallow') | 2 |
| | , | ('Black', '*', '*', 'ared', ' swallow', '*') | 10 |
| | Py | (' swallow', '*', ' Py') | 27 |
| | g | (' swallow', '*', '*', 'g') | 31 |
| | oc | (',', ' Py', 'g', 'oc') | 27 |

0.1    0.2

Figure 12: **Rule selection for a Wikipedia heldout sequence.** Analogous to Figure 5 but with optimal rule chosen from $\mathcal{R}_7^{\mathrm{all}}$ and with variational distance replaced with the $L^\infty$ metric for measuring distances between probability distributions. Model size: 1.4B.

| Predictions (transformer / rule) | Ground truth + Rule distance | Rule context | count |
|---|---|---|---|
| | video | (' video',) | 525296 |
| game | was | (' video', ' was') | 35709 |
| released | directed | (' video', ' directed') | 1837 |
| | by | (' was', ' directed', ' by') | 37813 |
| the | Michael | (' was', '*', '*', ' Michael') | 5779 |
| H , | **Sal** | (' Michael', ' Sal') | 240 |
| | omon | (' was', '*', ' by', '*', ' Sal', 'omon') | 66 |
| . , | **and** | (' was', ' directed', '*', '*', '*', '*', ' and') | 4339 |
| produced , | **prem** | ('omon', ' prem') | 2 |
| | iered | (' Michael', '*', '*', ' prem', '*') | 18 |
| | on | (' and', ' prem', '*', '*') | 7480 |
| | **C** | (' and', '*', ' on', ' C') | 334 |
| BC | MT | (' on', ' C', 'MT') | 720 |
| | on | ('iered', '*', ' C', '*', ' on') | 150 |
| October , | **February** | (' prem', '*', '*', '*', '*', ' on', ' February') | 226 |
| | | ('iered', ' on', ' February', '*') | 1429 |
| | 1 | (' on', ' February', ' ', '1') | 51089 |
| , | 5 | ('MT', '*', ' February', '*', '5') | 1 |
| | , | (' February', '1', ',') | 3 |
| | | (' on', ' February', '*', '1', ' ') | 8 |
| | 2 | (' February', '5', '*', '*', '2') | 2 |
| | 0 | ('1', '5', '*', '*', '2', '0') | 193674 |
| 1 | 0 | ('1', '5', '*', '*', '2', '*', '0') | 55855 |
| 9 | 6 | (',', ' ', '0') | 143834 |
| | . | (',', '2', '*', '0', '*', '.') | 4435 |
| \n | G | (' ', '2', '0', '*', '*', '.', ' G') | 2584 |
| aga , r | **AC** | ('0', '*', '.', '*', 'AC') | 86 |
| V , | **cut** | ('AC', ' cut') | 6 |
| | the | (' G', 'AC', '*', ' the') | 61 |
| video , | **ending** | ('6', '*', '*', 'AC', '*', ' the', '*') | 19 |
| | of | ('.', 'AC', '*', '*', '*', ' of') | 2 |
| | the | (' G', 'AC', '*', ' the') | 61 |
| video , | **video** | (' the', ' ending', '*', ' the', '*') | 2003 |
| , | out | (' cut', ' the', '*', '*', ' out') | 25 |
| of | because | (' video', '*', ' because') | 93 |
| it | of | (' out', ' because', ' of') | 689 |
| the | its | (' of', '*', ' video', '*', ' its') | 19 |
| sexual | suggestive | (' because', ' of', '*', ' suggestive') | 18 |
| nature , | **language** | (' because', '*', ' its', ' language') | 9 |
| . | Keith | (' of', ' Keith') | 1262 |
| ' | tells | (' because', '*') | 842884 |
| his , | **the** | (' of', ' suggestive', '*', '*', '*', '*') | 61 |
| audience , . | **audience** | (' Keith', ' tells', '*', '*') | 24 |
| to , | **,** | (' tells', ' the', ' audience', ',') | 16 |
| " | referring | (' the', '*', ',', ' referring') | 661 |
| | to | (' tells', '*', '*', ',', '*', ' to') | 29 |
| the | him | (' referring', '*', ' him') | 924 |
| as | shooting | (' referring', ' him', '*') | 23 |
| a | the | (' shooting', ' the') | 2570 |
| video , | **video** | (' to', ' him') | 114486 |
| . | as | (' to', ' him', ' as') | 6703 |

0.2  0.4

Figure 13: **Rule selection for a Wikipedia heldout sequence.** Analogous to Figure 5 but with optimal rule chosen from $\mathcal{R}_7^{\text{all}}$ and with variational distance replaced with the $L^\infty$ metric for measuring distances between probability distributions. Model size: 1.4B.

# E  Broader Impacts

Large language-models are having significant impacts on society, due to their use as question-answer tools and natural language generators. A better understanding of such language models will only serve to improve their capabilities. Our work here presents steps towards a fundamental understanding of language models, albeit in a small-scale regime far removed from those relevant for production systems. Given how far removed our work is from realistic datasets and use cases, we do not anticipate any direct negative broader impacts of our work.

