# OpenReview forum: "Understanding Transformers via N-Gram Statistics"
_NeurIPS.cc/2024/Conference — NeurIPS 2024 poster_

### Official Review · Reviewer_BciX · 2024-07-05

**Soundness:** 3
**Presentation:** 3
**Contribution:** 2
**Rating:** 6
**Confidence:** 4

**Summary:**

The paper studies how transformer-based large language models (LLMs) use context when predicting the next token by approximating these predictions with N-gram-based statistical rules. The authors propose a method to describe transformer predictions using simple N-gram rules and study how well these rules approximate the predictions. They claim their key findings include a new method to detect overfitting without a holdout set, insights into the progression from simple to complex statistical rules during training, a criterion for when transformer predictions align with N-gram rules, and an understanding of how well transformers can be approximated by increasingly complex N-gram rulesets. The experiments (primarily carried out on the TinyStories dataset and validated on Wikipedia data) seem provide possible insights into the statistical nature of LLM behavior.

**Strengths:**

- The proposed method for detecting overfitting without needing a holdout set is new and interesting for optimization applications.
- The paper includes visualizations and concrete examples.
- Although focused on specific datasets, the methods and insights could potentially be scaled to other domains.

**Weaknesses:**

- The study provides descriptive approximations without offering explanations into why certain rules work, not going deep enough in understanding the transformer behavior.
- The paper doesn't properly explore how fine-tuning on different datasets might affect the effectiveness of the proposed methods.
- The study mainly considers context lengths of up to seven tokens, which may not fully capture the long-range dependencies that transformers are capable of handling.

**Questions:**

- How do you plan to address the computational complexity associated with selecting the optimal N-gram rule for each context?
- Do you have any plans to transition from descriptive approximations to more explanatory models that can predict why certain rules work?
- Have you considered additional evaluation metrics that might capture other aspects of model performance and approximation quality?

**Limitations:**

- The computational overhead required to apply and test the N-gram rule sets at inference time could be significant.
- The proposed methods might not easily adapt to new or evolving datasets without substantial re-calibration or re-training, limiting their long-term applicability in dynamic environments
- While N-gram rules provide a way to approximate transformer behavior, the results might not always be easily interpretable by humans.

---

> ### Author Rebuttal · Authors · 2024-08-05
>
> We thank the reviewer for their time and concerns.
>
> *Weaknesses*
>
> 1. “The study provides descriptive approximations without offering explanations into why certain rules work, not going deep enough in understanding the transformer behavior.”
>
> While it is true we do not provide explanations (as explicitly mentioned in the paper), we believe our work provides a novel complement to such efforts. For instance, there is a growing body of work to explain very particular LLM functions, say using circuits (e.g. Indirect-Object-Identification or copying). It may be difficult, if not impossible, to try to explain the myriad of heterogeneous behaviors arising from n-gram templates, each with their own mechanisms, within the scope of a single paper. Rather the perspective of our paper is to give a broad analysis in terms of description (what we call “form” in the introduction) that later can be used to guide research into explanation (what we call “selection” in the introduction).
>
> In other words, before looking for explanations, it is useful to have a rich set of descriptions (as much as is feasible) first. Example: it may be useful to describe the possible different weather patterns before developing the underlying physics that explains the different weather patterns.
>
> 2. “The paper doesn't properly explore how fine-tuning on different datasets might affect the effectiveness of the proposed methods.”
>
> This is true but we believe that is the scope for followup work. It would muddy the waters of n-gram analysis to have multiple datasets and sequential training, since there would be ambiguity about how to combine n-gram statistics from multiple datasets. In fact, we think a very interesting followup paper would be precisely to explore such questions (e.g. when fine-tuning, are n-gram statistics of the new dataset used to override old ones or is some mixture of statistics happening?). But for laying groundwork analysis, it is most appropriate to focus on a single dataset.
>
> 3. “The study mainly considers context lengths of up to seven tokens, which may not fully capture the long-range dependencies that transformers are capable of handling.”
>
> With more engineering and compute it is true we could tabulate n-grams with larger n. And by definition, we could get better approximation because we have more n-grams available. But we believe we already have shown good approximation by n-grams rules for up 7 tokens of context: is there anything the reviewer wishes to see beyond guaranteed improved approximation?
>
> *Questions*
>
> 1. “How do you plan to address the computational complexity associated with selecting the optimal N-gram rule for each context?”
>
> For suffix based N-gram rules, we discovered after setting up our N-gram database that there is a suffix array structure which makes querying N-grams very fast and scalable, e.g. an N-gram querying API is available for datasets of order 10^8-10^9 tokens (https://huggingface.co/spaces/liujch1998/infini-gram). This data structure can be used to scalably find optimal N-gram rules for the suffix rules. For subgram rules and marginal rules, some other clever data structure may enable quick N-gram querying but we have not invested effort into this direction yet.
>
> 2. “Do you have any plans to transition from descriptive approximations to more explanatory models that can predict why certain rules work?”
>
> Yes but that is not within the scope of the present paper. The situation is analogous to the Chinese Room Argument: we are able to describe the form of the outputs but have no claim on the inner workings. We believe followup work using mechanistic interpretability tools could address the problem of rule usage/selection. Nevertheless, we want to stress that one of our remarkable results is to be describe predictions in terms of n-gram rules as often as we can (78% in a certain sense on TinyStories).
>
> 3. “Have you considered additional evaluation metrics that might capture other aspects of model performance and approximation quality?”
>
> As this is a very open-ended question which we are uncertain how to answer, but we provide two details here which might be useful to the reviewer:
>
> We varied the choice of metric to choose the optimal rule by choosing the $L^\infty$-norm for Wikipedia. The same qualitative picture emerges and we saw no drastic change in the numbers.
>
> We also tried forming some simple predictive models using the rules (e.g. using simple statistics to decide which to use) but were not successful. This is why we limit ourselves to “descriptions” since selection/”explanation” is difficult. Nevertheless, prior to our work, it was not even clear or quantified to what extent n-gram rules describe/approximate LLM predictions.
>
> *Limitations*
>
> 1. “The computational overhead required to apply and test the N-gram rule sets at inference time could be significant.”
>
> Yes, though for suffix rules, the method is known to be scalable, see above.
>
> 2. “The proposed methods might not easily adapt to new or evolving datasets without substantial re-calibration or re-training, limiting their long-term applicability in dynamic environments”.
>
> We’re unsure what the baseline here. Isn’t the above a weakness for nearly the whole of supervised learning? When a new dataset arrives, one almost always has to retrain or do some kind of adaptation accordingly.
>
> 3. “While N-gram rules provide a way to approximate transformer behavior, the results might not always be easily interpretable by humans.”
>
> N-gram models are among the simplest, interpretable models (since they are based on simple frequency-based statistics). Does the reviewer not find the descriptions provided in the “Rule context” column in figures such as Figure 5 rather interpretable?

---

> > ### Comment · Reviewer_BciX · 2024-08-10
> > **Thank you for your response.**
> >
> > I agree that N-gram models are inherently simple and interpretable due to their reliance on frequency-based statistics; however, my concern is that while individual N-grams are indeed easy to understand, the challenge arises when these N-grams are aggregated and interact within a complex model like a transformer.
> > Anyway, I appreciate the time you spent writing this respose.

---

### Official Review · Reviewer_H4bx · 2024-07-11

**Soundness:** 2
**Presentation:** 2
**Contribution:** 2
**Rating:** 5
**Confidence:** 3

**Summary:**

The authors use n-gram statistics and regular expression templates to study how well they describe the predictions of Transformers-based models. They craft rules that vary in context length and/or the number of marginalised context variables to predict the next word. They use their framework to study overfitting (and memorisation) of increasingly complex template rules. They discover a counter-intuitive phenomenon: longer n-gram rules can be less predictive of a model's behaviour than shorter; an insight they name model-variance trade-off.

**Strengths:**

The idea of defining templates that apply both to the data distribution and the model's generation is good and sound.
I appreciate the idea of having a mathematical methodology as it states clearly what is the objective and how rules are defined (despite a few things I do not fully understand and I reckon may be wrong; see next sections).
Findings are noticeable: 7-gram rules better approximate a model's behaviour than 8-grams (but please refer to the next section for. a caveat on this finding).
The model-variance explanation of why some rules are better than others is intriguing (but see weaknesses).

**Weaknesses:**

I am concerned about the evaluation, in particular Eq. 10. You measure how closely the dataset adheres to the template rules and compare it to your model’s behaviour. With such a small dataset, the fact 7-gram rules better approximate the distribution of data may be caused by a considerable drop in the overall frequency (and thus, variance) of 8-grams (vs. 7-grams). See a question about that.
In summary, my biggest concern is whether the best-rule is identified by the model-variance trade-off or the intrinsic larger variance of 8- vs. 7-gram rules (i.e., 8-grams are more difficult to predict because you don't have enough data, so the 7-gram is optimal). With larger datasets and different models, results can vary.

In fact, another major concern is that you only have two datasets and one model to prove your hypothesis.
Furthermore, it is not clear if you take inspiration from Chinchilla's architecture (it seems to me you do not use a pre-trained model). Otherwise, it would be nice to test this hypothesis with smaller, purely next-word-prediction trained models (even gpt-2 or Pythia would be good).

Other concerns (in descending order of importance):
Figure 2: low R-squared may mean the regression failed and not lack of "correlation". Furthermore, you mention 7-grams but some plots are for 8grams (see y-axis of figs (a,b)). While I understand why, the plot is named 7-grams, so I was a bit confused at the beginning. It's better to show less in this case or make it clear also in the caption that the top-plots refer to 8grams.

(minor) Equation 10 seems to me wrong: p and q are distributions, you can’t subtract them (unless i stands for p(t=v_i |C), but that has to be clarified).
If not, probably you want to say something like Divergence(p || q). On a side note, I think this is one of the cases where an asymmetric rule makes sense. You want to measure the amount of “information” a model lacks to turn p (its prediction) into q (the ground truth).

(minor) I am a bit confused by the notation in Section 4 (see questions, though the general idea is clear enough).

**Questions:**

In order of importance:
(major) Can you give a clear definition of model-variance in terms of predictive capabilities of the model and when compared to an n-gram rule?

(major) Have you checked if there is a large drop in the variance of 7 vs. 8 grams in your dataset? I suspect that causes the 7-gram rules to be optimal and not the model-variance trade-off. In other words, the variance of 8-grams in your dataset is so large (because you don't have enough data), that the architecture you use cannot approximate it.

(minor) What if the optimal rule you find only correlates with a model's behaviour (e.g., due to the size of the dataset)? Can you give some evidence that a model is actually using n-gram rules to predict the next token?

(minor) Eq. 10: what is i? Is p(t=v_i | C), with v_i the i-th token in the vocabulary, or the i-th run of the model?

(minor) Line 111, the expression +-*+ should produce C_{-5}*C_{-1} and not C_{-4}*C_{-1}, or am I missing something in your notation? Is it for the left padding you mention earlier?

(minor) Eq. 6 (left hand side): why don’t you have C_{-4} at the denominator?

**Limitations:**

See previous points.

---

> ### Author Rebuttal · Authors · 2024-08-05
>
> We thank the reviewer for their time in reviewing our work and appreciating the soundness of using n-gram statistics to understand LLMs.
>
> Regarding the 8-gram model vs rules $R_7^{suffix}$ or more generally $R_7$ using up to 7 tokens of context: We are not entirely sure we fully understand the reviewers concern, but we provide some details which we hope addresses our interpretation of them:
>
> a) Note that the 8-gram model (which uses all 7 of the most recent tokens of context, when they occur in the training data) is one rule among the many in the ruleset $R_7^{suffix}$. Hence by definition, $R_7^{suffix}$ outperforms an 8-gram model because it has many more templates with which to optimize when comparing with an LLM next-token prediction.
>
> b) For our Figure 2 involving 7 tokens of context, we believe the reviewer is asking whether there is some simple phenomenon like having high count k-grams in the context that is “causing” the LLM to rely on such k-grams (k < 8) instead of the 8-gram model when the count of corresponding 8-grams is low.
>
> This is certainly not the case! Please see Fig 2 in the Author Rebuttal pdf. Especially in Fig 2(b) we see that across all settings in which the optimal rule is a k-gram rule 2 <= k <= 8, there are always 7-grams in which the corresponding 8-gram model has several thousand instances (all 7 curves in Fig(b) have maximum value at least several thousand). Likewise Fig 2(a) suggests there isn't a simple count-based reason for when the 8-gram rule is optimal versus the 7-gram rule is optimal (the corresponding pink/brown curves are qualitatively similar, showing that the associated n-gram distributions are similar for both cases, 2 <=n <= 8).
>
> In fact, the point of our Approximation-Variance relation is that we failed to find any simple count-based method of selecting optimal n-gram rules (of the type the reviewer is suggesting). The lesson is that knowing *which* rule is selected is hard, but that knowing *some* good rule exists can be more easily guaranteed (by Approximation-Variance, when the context has model prediction with low variance). This important finding can be emphasized in the revision. (See also Section C.3 and Figure 8 for related insights.)
>
> “Can you give a clear definition of model-variance in terms of predictive capabilities of the model and when compared to an n-gram rule?”
>
> If the question is whether we know of a way to characterize model variance other than its definition, then the answer is we do not know - it is a genuine mystery! Contexts which occur only once (so next token distribution has zero entropy) can have both high and low model variance (see the large vertical spread in Fig 2(c) at count = 1). It is an open problem to be able to explain such wide-ranging behavior. (See also comments in general Author Rebuttal.)
>
> This question/puzzle can be emphasized as a future direction of work.
>
> Regarding choice of models, observe that GPT-2 is 1.5B and does not have a public dataset (rendering n-gram analysis unavailable). While Pythia's dataset is available, it is very large for our current n-gram analysis. Our model sizes range from 160M to 1.4B (within the range of the smaller Pythia family). Thus our models are within the "smaller models" range the reviewer suggested?
>
> *Minor questions / notation*:
>
> Fig 2: The y-axis is correctly labeled 8-gram because we are comparing the *8-gram rule* prediction with the transformer prediction on a *7-gram context*. (This is a constant off-by-one source of headache with n-grams: the n-gram rule uses n-1 tokens of context)
>
> “What if the optimal rule you find only correlates with a model's behavior (e.g., due to the size of the dataset)? Can you give some evidence that a model is actually using n-gram rules to predict the next token?”
>
> Could the reviewer please clarify what is meant by “only correlates with a model’s behavior” mean? A rule can describe a model’s behavior in that it provides a predictive probability distribution that is close to the model’s predictive probability distribution.
>
> For the second question, as noted, we only use rules to describe predictions, not explain them. Our analysis is a black-box analysis studying the “form” of prediction, not one that uses model internals to confirm that they “use” (i.e. “select”) such rules (this is the “form vs selection” distinction in the paper introduction). It is analogous to a Chinese Room Argument situation: we are able to describe the form of the outputs but have no claim on the inner workings. We believe followup work using mechanistic interpretability tools could address the problem of rule usage/selection. Nevertheless, we want to stress that part of the our novelty results is to be describe predictions in terms of n-gram rules as often as we can (78% in a certain sense on TinyStories).
>
> “Eq. 10: what is i? Is p(t=v_i | C), with v_i the i-th token in the vocabulary, or the i-th run of the model?”
>
> It is p(t=v_i | C), with v_i the i-th token in the vocabulary. The definition of variational distance in Eq. 10 involves the distance between two probability distributions regarded as vectors indexed by i. So for probability vectors arising from language models, i indexes the vocabulary. And so yes, we can subtract probability distributions (because they are vectors and we can subtract vectors).
>
> “Line 111, the expression +-*+ should produce C_{-5}*C_{-1} and not C_{-4}*C_{-1}, or am I missing something in your notation? Is it for the left padding you mention earlier?”
>
> The line there is correct. Parsing +-*+ from right to left, the “+” keeps C_{-1}, the * yields a *, the - drops the C_{-3}, and the final + keeps the C_{-4}. Padding means we drop all remaining tokens to the left (hence we drop C_{-5}).
>
> “q. 6 (left hand side): why don’t you have C_{-4} at the denominator?”
>
> Apologies, this was a bad typo! There should absolutely be a C_{-4} in the denominator (just like on the right hand side).

---

> ### Comment · Area_Chair_XENr · 2024-08-12
> **Reviewer, please respond to authors' rebuttal**
>
> Hello Reviewer,
>
> Please take a moment to read and acknowledge the authors' rebuttal. Especially considering you gave a "borderline" review score, it would be helpful if you could weigh in on whether their response pushes you one direction or the other.
>
> Thanks,
>
> AC

---

### Official Review · Reviewer_GxTz · 2024-07-13

**Soundness:** 2
**Presentation:** 3
**Contribution:** 3
**Rating:** 7
**Confidence:** 3

**Summary:**

The authors use 160M parameter models trained on the TinyStories dataset (artificially generated short stories, 480M tokens, made up of "vocabulary consisting of about 1500 basic words") to study transformers by comparing their predictions to the predictions of n-gram models.

This leads to a few observations [quoted from the paper here]:
1. (Approximation-Variance Tradeoff) We observe an “approximation-variance tradeoff",
which roughly states that next token LLM predictions that have low variance (across
different training runs) tend to be well-approximated by N-gram rules. (Section 5)
2. (Curriculum Learning Dynamics) By grouping our N-gram rulesets in terms of complexity
(as measured by the amount of context they use), we discover the various ways in which the
learning dynamics of LLMs implement a statistical type of curriculum learning, in which
easier rules are eventually supplanted by more complex ones. (Section 6.1)
3. (Overfitting Criterion) Based on our analysis of approximating LLM predictions by N-gram
rules, we propose a simple and novel procedure for detecting overfitting of LLMs during
training. The procedure makes no use of holdout data and it makes quantatively precise the
intuition that overfitting corresponds to a model memorizing long context at the expense
being able to generalize through making use of subcontext. (Section 6.2)
4. (Approximation Strength) We study how well LLM predictions can be approximated by our
N-gram rulesets, noting that significant gains in top1-accuracy occur as we increase ruleset
complexity and diversity. We also visually ground these approximations with concrete
examples (Figure 5), which may form the basis for dataset attribution methods in future
work. (Section 7)

**Strengths:**

LMs are everywhere but we don't really know how they work and how the learn. Figuring this out might lead to stronger, more efficient models.
I usually don't like interpretability research because most methods employed just don't make sense-  but I *love* the approach taken by this paper of using n-gram models to analyze transformer models. These results seem straightforward but I'm pretty sure that this paper is the first to ever present them.

**Weaknesses:**

There's a huge weakness in this paper- they analyze extremely small (160M param) models trained on a total toy dataset (TinyStories). TinyStories is a very small dataset made up of short stories that were generated by an LM. The vocabulary in this dataset is limited to 1.5k basic words.

All of these things together mean that none of these results might transfer towards bigger, realistic LMs like LLaMA 3 70B. If this paper would have been written with LLaMA 3 70B as the model being used I would have argued for a strong accept, but because the models used here are so tiny I'm less excited about this contribution.

Second thing- the authors put out a lot of what seems like 'intermediate analysis'- observations they noticed, but they don't explain what the conclusion from that observation is or why it's interesting. For example fig 2.

Third- why is the 'Overfitting Criterion' interesting? Is anyone overfitting their LMs? Would anyone need this in practice?

**Questions:**

1. is the Approximation-Variance Tradeoff true also for predictions that have low entropy?

---

> ### Author Rebuttal · Authors · 2024-08-06
>
> We thank the reviewer for their time in reviewing our work and appreciating its novelty.
>
> *Scale*
>
> The reviewer noted we only trained a small 160M model on TinyStories. Perhaps the reviewer overlooked that we also trained on Wikipedia using 1.4B models?
>
> LLaMa 3 70B would not have been an appropriate model to study for several reasons. First, the underlying training data is not public, so it is impossible to perform the n-gram analysis of our paper. Second, SOTA LLMs involve both a pretraining phase and various additional training steps on top. Our paper only focuses on the single pre-training step for an initial controlled study. Finally, our new results (see below) suggests scaling past to 70B might not be so informative given the current context length available for our n-grams.
>
> TinyStories under our tokenizer has 23K distinct unigrams (Fig 7) and so is not as trivial a dataset as suggested. Moreover, the simplicity of TinyStories can be regarded as a feature not a bug: we want a dataset whose “effective” context length is small when being probed by n-gram statistics (with n small due to computational limitations). A children’s story will intuitively have less context dependence than a Wikipedia article, since the latter may rely on facts much earlier in an article. The two extremes given by simplistic language for TinyStories on the one hand and high-quality language via Wikipedia on the other hand, we believe, are the appropriate extremes to consider for the initial validation of our work.
>
> The reviewer’s question about scale prompted us to look into past experiments we did long ago on various model sizes which we can include in the paper revision (actual numbers will differ slightly since these older experiments had different train settings).
>
> Experiment 1: TinyStories
>
> | Model Size | Top-1 Acc (Ground Truth) | Top-1 Acc (Optimal Rule from $R_7$) | Optimal Distance (from $R_7$) |
> |---|---|---|---|
> | 160M | 68.5 | 79.0 | 0.164 |
> | 420M | 69.4 | 79.1 | 0.169 |
> | 1.4B | 70.3 | 79.2 | 0.170 |
>
> Experiment 2: Wikipedia
>
> | Model Size | Top-1 Acc (Ground Truth) | Top-1 Acc (Optimal Rule from $R_6$) | Optimal Distance (from $R_6$) |
> |---|---|---|---|
> | 160M | 51.5 | 63.6 | 0.163 |
> | 420M | 53.7 |63.9 | 0.171 |
> | 1.4B | 55.8 | 63.6 | 0.181 |
>
> These results show larger models improve in performance (ground truth acc increasing on validation set), but they slowly depart in approximation from our n-gram rules (the distance to the rules slowly increases while the top-1 acc of the optimal rule plateaus). We conjecture this is happening because larger models are better able to make use of longer context when appropriate: this leads to noted increase in distance of predictions from rules using 6-7 tokens of context without impacting the top-1 acc between the rules too much.
>
> Nevertheless, the stability of our n-gram results when scaling up the model size suggests that our results hold at scale and that we expect to see similar results across a wide range of models on large datasets.
>
> *Overfitting*
>
> Whether overfitting occurs in practice is, we believe, the incorrect perspective to take since it would, e.g., dismiss phenomena such as grokking. The significance of our overfitting criterion is the insight it gives into understanding generalization vs memorization (just like with grokking), which is a subject often discussed with informal intuitions instead of precise quantification. The fact that our overfitting criterion quantifies generalization vs memorization in a way that only uses the training set and has a simple explanation in terms of n-gram statistics is, in our opinion, a novel discovery.
>
> *Approximation-Variance (AV) for Low Entropy*
>
> The AV phenomenon is also true for predictions that have low entropy! In particular, consider n-gram contexts that occur only once in the training data (so zero entropy for next-token distribution). We replot Fig 8(a,c), using only unique full-context (those starting with BOS) bigram contexts as Fig1(a,b) in the Author Rebuttal. Fig 1(a) shows how for such contexts, aside from some outliers, those with low model variance (x-axis) will have good rule approximation (low y-value) and vice versa. On the other hand Fig 1(b) shows that a frequency based analysis of which contexts lead to low model variance (needed for good n-gram approximation) leads to a poor fit. (Here the only nontrivial count available is the number of occurrences of the last token of the context and it can be very large and still lead to high model variance.)
>
> To summarize, one significance of our AV result is that an n-gram context being rare (even unique) is not predictive of the model prediction deviating from the associated n+1-gram rule. Rather it’s the variance of the predictive distribution associated to the n-gram context which is correlated with approximation by the n+1-gram rule (and this variance can surprisingly be low even for count = 1). This is quite surprising and future work would be to understand what makes some unique n-grams have low model variance while others have high ones. This can be emphasized in the paper revision.
>
> *Intermediate Analysis*
>
> The reviewer noted that there is “intermediate analysis” whose significance is not clear (e.g. Fig 2). We hope our answer to the reviewer’s question above partially addresses this concern. To summarize, Fig 2 and the AV results of Section 5 is meant to answer the question “When does a context result in an LLM prediction well-approximated by n-gram rules?” (Answer: Often those with low variance. Simple count based statistics will be much less predictive since rare contexts can be approximated while common ones may not well-approximated alike) whereas Sec 7 measures how often this occurs (78% on TinyStories). Sec 6 provides additional insights into how n-gram rules are being used during training / overfitting. We hope this makes the structure of the paper more clear and will make these points more explicit in the paper revision.

---

> > ### Comment · Reviewer_GxTz · 2024-08-12
> >
> > changed my score to 7.
> >
> > regarding the comment about llama 3.1: you're right, i understand now why you cant show results on that (you don't have the train set).
> >
> > regarding the 2 tables with experiments on 420M and 1B param models:  that looks great, please add it to the paper.

---

> ### Comment · Area_Chair_XENr · 2024-08-12
> **Reviewer, please respond to authors' rebuttal**
>
> Hello reviewer,
>
> Please take a moment to read and acknowledge the authors' response.
>
> Thanks,
>
> AC

---

### Author Rebuttal · Authors · 2024-08-06

We thank all the reviewers for their valuable feedback. Attached is a pdf of Figures 1 and 2 relevant to the individual author rebuttals. Some high level comments to reiterate some overlapping feedback:

1) We have substantial evidence that our results hold with scale. This is justified via additional experiments noted in the response to Reviewer #1.
2) A key message from our Approximation-Variance result is that simple count/entropy-based criteria for which contexts lead to LLM predictions being well-approximated by n-gram rules do *not* seem to be readily available (this is what Fig 1 and 2 in the rebuttal pdf show along with the responses to Reviewer #1 and #2, in addition to Fig 2 and 8 of the main paper). Indeed after much tinkering, the main correlation we identified for being well-approximated by n-gram rules (as measured by low optimal distance to rules) was having low model variance. It is an interesting open question as to whether this latter property has some simpler characterization and we will leave this to future work. (This is intuitively a difficult problem to address, and is closely related to other works that study which training examples are hard or end up being memorized/forgotten by neural networks, e.g. [1] and [2]. To the author's knowledge, there is no systematic understanding of which training examples are hard / easily memorized.). To reiterate, such difficulties is why our paper takes a descriptive approach to n-gram rules rather than explanatory one, because simple hand crafted features (like counts, entropy) which would provide simple explanations do not seem to go very far.

[1] Toneva et al. An Empirical Study of Example Forgetting during Deep Neural Network Learning. ICLR 2019.

[2] Carlini et al. Quantifying Memorization Across Neural Language Models. ICLR 2023.

---

### Decision · Program_Chairs · 2024-09-25

**Decision:**

Accept (poster)

**Comment:**

SUMMARY

How well do n-gram rules from statistical language models approximate the predictions of transformer-based neural language models when they are built/trained off the same dataset? The paper shows that high-confidence LLM predictions tend to be well-approximated by n-gram rules and that LLMs learn simple n-gram rules before they learn more complex ones. They further use their methods to detect when an LLM has overfitted to the training data.

REASONS TO ACCEPT

Reviewers really like how the authors use classical methods from NLP  to try and understand modern LLMs' behaviours. The results are intriguing, and there are natural extensions for impactful future work.

REASONS TO REJECT

The "LLM" used by the paper is actually tiny: 160M parameters. It is unclear whether the results will be the same/different if the methods were evaluated on a much larger language model trained on a much larger dataset. It is also unclear how the methods in the paper could be efficiently scaled up to this more realistic "large" setting.

CONCLUSION

Overall, the ideas in this paper are worth sharing at NeurIPS, and I look forward to future work which addresses the scaling challenge.